neuroscience/psychology/cognition

threat learning, representational similarity analysis, intention, fMRI, social learning

**Author for correspondence:**
Irem Undeger
e-mail: irem.undeger@ki.se

# Model-based representational similarity analysis of blood-oxygen-level-dependent fMRI captures threat learning in social interactions

Irem Undeger[1], Renée M. Visser[3], Nina Becker[1],
Lieke de Boer[2,4], Armita Golkar[1,5] and Andreas Olsson[1]

[1]Section for Psychology, Department of Clinical Neuroscience, and [2]Department of Neuroscience, Care Sciences and Society, Karolinska Institutet, Nobels väg 9, 171 77 Stockholm, Sweden
[3]Department of Clinical Psychology, University of Amsterdam, Nieuwe Achtergracht 129-B, 1018 WT Amsterdam, The Netherlands
[4]Aging Research Center, Tomtebodavägen 18A, 11330 Solna, Stockholms Lân, Sweden
[5]Department of Psychology, Stockholm University, Frescati Hagväg 14, 114 19, Stockholm, Sweden

IU, 0000-0002-7018-5077

Past research has shown that attributions of intentions to other's actions determine how we experience these actions and their consequences. Yet, it is unknown how such attributions affect our learning and memory. Addressing this question, we combined neuroimaging with an interactive threat learning paradigm in which two interaction partners (confederates) made choices that had either threatening (shock) or safe (no shock) consequences for the participants. Importantly, participants were led to believe that one partner intentionally caused the delivery of shock, whereas the other did not (i.e. unintentional partner). Following intentional versus unintentional shocks, participants reported an inflated number of shocks and a greater increase in anger and vengeance. We applied a model-based representational similarity analysis to blood-oxygen-level-dependent (BOLD)-MRI patterns during learning. Surprisingly, we did not find any effects of intentionality. The threat value of actions, however, was represented as a trial-by-trial increase in representational similarity in the insula and the inferior frontal gyrus. Our findings illustrate how neural pattern formation can be used to study a complex interaction.

# 1. Introduction

Many of our prominent memories involve social interactions; the childhood memory of an embarrassing incident at school that was followed by snarky comments, or a recent birthday party where everyone was so kind. The memories of these events, and of the individuals in them, are shaped by both the consequences of these individuals' actions and our knowledge about the intentions behind them. When we interpret another person's negative actions, we commonly take into account their agency. Previous research has shown that intentional actions leading to threat outcomes lead to the formation of distinct neural signatures throughout the cortex [1]. Here, we aim to replicate these findings in a novel dataset and expand on this research by disentangling neural signatures of key features (i.e. agency of others and the threat value of their actions) of social interactions as they evolve over time. To this end, we used a representational similarity analysis (RSA) approach to formulate specific predictions regarding the development of neural signatures related to threat learning as a social interaction took place. Additionally, we investigated the update of these signatures in a re-encounter with individuals that previously caused harm using an extinction learning phase.

Past research has shown that an intentionally harmful action is rated as 'deserving punishment', and the individual is blamed, whereas an accidental action can be forgiven [2–4]. In our recent study [1], in which participants were delivered electrical shocks by two confederates following a staged social interaction, intentional shocks were perceived as more uncomfortable, and the intentional confederate less likeable. Neuroimaging data from functional magnetic resonance imaging (fMRI) showed that the intentional delivery of shocks was represented as the formation of a distinct neural signature throughout the cortex, including in the anterior cingulate cortex (ACC), the inferior frontal gyrus (IFG) and most prominently in the insula. These neural signatures involved an increase in correlations between neural activation patterns related to consecutive trials of the threat stimulus with the 'intentional' confederate, relative to the 'non-intentional' confederate and to non-threat stimuli. This specific trial-by-trial application of RSA allowed the investigation of similarities between neural activation patterns throughout the experiment. Here, we present a replication study.

Because learning during a social interaction involves integrating the intentionality and the outcomes of a social interaction partner's actions in time, it is important to be able to separate different processes that might govern each feature of the interaction. Two such key features are: (i) the social information about the interaction partner, and (ii) the outcome of the partner's action. To accomplish this, we combined the trial-by-trial RSA method with a regression-based analysis adopted from research on non-human primates [5]. In our application of this method, we used regressors that model the formation of neural activation patterns as information is gathered and processed. Instead of taking a correlational value between experimental data and a single template that represent theoretical pattern correlations for one feature of the social interaction, in this method, multiple templates are entered as predictive weights in a multiple regression analysis on the similarity matrix. This means that using a single statistical test, we can assess the weight of different features of the social interaction that lead to the formation of neural patterns, fitting this at the level of individual participants. Our models consisted of (i) a threat learning regressor that models neural pattern formation due to threat learning from the outcomes of the social partners actions, regardless of intentionality, (ii) an intentionality regressor that models neural pattern formation based on the intentionality of an action regardless of its outcome, and (iii) regressors that model the integration of intentionality or unintentionality and the outcome of the action as learning unfolds (figure 1*b*). To understand if the 'learned' patterns update in a future encounter, we added an 'extinction learning' phase during which participants underwent the same task, but without experiencing any threat outcomes. This additional phase of the experiment allowed us to investigate not only the neural signatures of learning during a social interaction, but also if these signatures are resistant to an update at a later point. Indeed, previous research has demonstrated a resistance to extinction (i.e. safety learning) of learned threat responses to certain categories of social stimuli, such as faces of out-group individuals [6,7].

In our task, participants received electrical shocks as a consequence of the confederates' choices. We analysed the time-course of neural pattern formation by implementing the trial-by-trial RSA method, and used a regression-based analysis to disentangle the weight of threat learning from the weight of intentionality behind the threatening actions. We selected regions of interest (ROI) based on previous research: the ACC, the amygdala, the insula, hippocampus and the ventromedial prefrontal cortex (vmPFC) were selected based on their role in threat learning [8] and the left and right temporoparietal junction (TPJ), dorsomedial prefrontal cortex (dmPFC), and right and left superior temporal sulcus (STS) were selected based on their role in intentionality processing and theory of mind (ToM) [9,10].

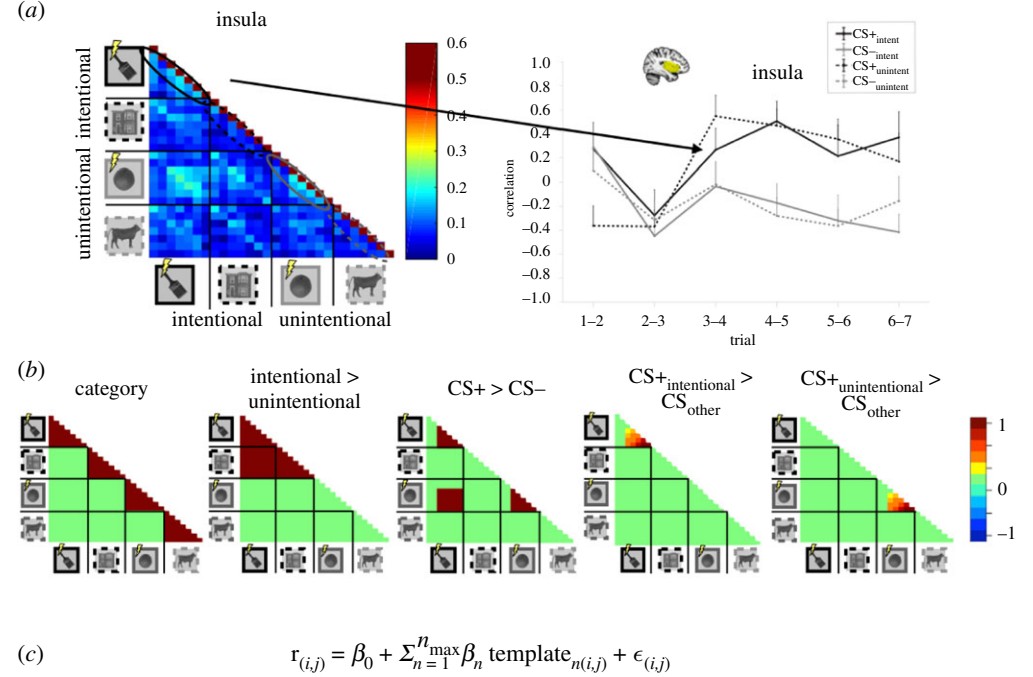

**Figure 1.** The RSA regression method. (*a*) The $28 \times 28$ RSM for the insula ($n = 26$), and trial-to-trial correlation values for consecutive trials. (*b*) Template regression matrices used for the chosen option period. Here, instead of accounting only for the consecutive trials as done in (*a*), template regression matrices were created to account for similarity between trials that are non-consecutive. As seen in the $CS+_{intentional} > CS_{other}$ template, an increase in correlations was modelled between consecutive trials (off diagonal line) and the rest of the quadrant for the intentional CS+ choices (off diagonal values). (*c*) The regressor equation used to compute RSA matrix similarity to the regressor templates. Here, templates from (*b*) are entered into the regressor to assess their individual weights. This allows us to not only test individual contributions of each template but also their relative weights to each other. Compared with the method used in section (*a*), this allows for both investigating the off-quadrant values that represent non-consecutive trials, and also to statistically observe the weight each carries.

In order to assess if the intentionality manipulation was successful, we used post-experimental questionnaires, which included questions about the social interaction partners and the interaction itself (electronic supplementary material, S2). To assess successful threat learning, we measured both contingency awareness (electronic supplementary material, S3) and pupil dilation.

Behaviourally, based on previous research, we expected participants to learn the contingencies between stimuli and the threat outcome (shock) as indicated by both explicit expectancy ratings [11] and pupillary dilation (physiological arousal) [12]. We also expected participants to report greater levels of discomfort [13,14], stronger feelings of revenge [10] and an inflated number of expected and received shock from intentional versus unintentional interaction partners [12].

On the neural level, we aimed to identify brain regions that are able to: (i) detect intentional versus unintentional social interactions (Intentional > Unintentional), (ii) detect threat expectation for conditioned stimuli (CS+ > CS−), (iii) detect threat expectation only in the intentional context and with a gradual learning effect, as opposed to an unintentional one ($CS+_{intentional} > CS_{other}$ versus $CS+_{unintentional} > CS_{other}$). Our main hypothesis about patterns of neural activity during the threat learning phase was that the intentionality of threatening actions would be represented in the insula, the ACC and the IFG [1]. Furthermore, we hypothesized that threat learning, regardless of intentionality, would be represented in the amygdala, the hippocampus, the ACC, the insula and the vmPFC [1,15,16]. For the extinction learning phase, our hypotheses were threefold; (i) if the decrease in learned fear responses is indeed represented by a decrease in neural pattern similarity over time, we would observe this change in regions that capture learning; (ii) if the extinction learning process leads to a 'resistance' in neural pattern correlations (i.e. no decrease) for intentional threatening choices, this would be captured in the regions that were involved in both processing intentionality and threat learning; and finally (iii) if certain regions support the formation of an inhibitory memory trace (e.g. vmPFC [17]), we would expect to see an increase in neural pattern similarity, similar to what was observed during threat learning.

# 2. Material and methods

## 2.1. Subjects

A power analysis conducted prior to the MR experiment indicated that to replicate the effect of threat learning from pupil dilation responses of Visser *et al.* [8] ($\eta^2 = 0.21$) with 95% power, and $\alpha = 0.05$, a sample size of 23 would be required. Considering exclusion criteria mentioned below (e.g. substantial head motion), 31 healthy, right-handed individuals with normal or corrected-to-normal vision were recruited via flyers and online recruitment system. Data were discarded from the analysis of fMRI data if the participant had substantial head motion (greater than 2 mm in any direction) ($n = 4$, for each of the two phases of the experiment). Data from one participant had to be discarded due to technical difficulties in the scanner. The final sample of the fMRI analysis for the learning phase included 26 participants (eight males, all right-handed) between 19 and 40 years of age (mean: $24.54 \pm 5.16$). As an independent index of threat learning we collected pupillometry data, excluding participants who had more than 33% of trials of any experimental condition (i.e. CS+$_{intent}$, CS+$_{unintent}$, CS−$_{intent}$ and CS−$_{unintent}$) missing. Missing trial was defined as greater than 50% missing sample for that trial ($n = 3$) [1,18]. Thus, fMRI data are reported for 26 participants and pupillometry data are reported for 23 participants (out of the same sample). All participants gave their written informed consent before participation and were naive to the purpose of the experiment. Two confederates were recruited from the city of Stockholm for the experimental manipulation, each assigned to the intentional or unintentional role in a counterbalanced manner. The procedures were executed in compliance with relevant laws and institutional guidelines and were approved by the local ethical board (dnr: 2017/138-31/2).

## 2.2. Stimuli

Greyscale photographs of the confederates served as face stimuli. The four CS images were chosen from four different categories: animals, fruits, tools and buildings. The images were obtained from the website www.lifeonwhite.com and from publicly available resources on the Internet, with their background removed [19]. These categories were chosen as they are represented in different regions of the brain [20] in order to minimize generalization effects between the CSs during learning [21]. All stimuli were equalized to match luminance using the SHINE Toolbox [22], in order to eliminate the effect of luminance in the pupillary responses.

## 2.3. Experimental procedure

Participants met the confederates immediately upon arrival at the experiment. The same individuals acted as confederates for all participants. Both the participant and the confederates were briefed together before the experiment began. The researcher explained the different roles in the experiment with a set script (electronic supplementary material, S1).

Using a lottery containing small pieces of paper that indicated 'MR scanner' or 'Outside', the participant was led to believe that only they chose the paper that indicated 'MR scanner' and that the confederates drew papers that indicated 'Outside'. Upon revealing their own ballots, the confederates left the room. At this point, the participant repeated the instructions given during the initial briefing to the researcher to make sure the participant understood their role. Only when alone, the participant learned that even if confederates chose not to deliver electrical shocks, the participant would still receive them. This allowed us to manipulate the intentionality of the shocks, as the participant believed that the unintentional confederate had no knowledge of shocks being delivered. Hence, equal number of shocks was delivered by both confederates without compromising the believability of the task.

We used a modified threat learning paradigm in which the two social interaction partners (confederates) delivered shocks to the participant by choosing one out of two images (figure 2*b*). The participant was led to believe in the presence of an online screen sharing system which shows choices made by each interaction partner in real time using a screen sharing software (figure 2*a*). At the beginning of the experiment, the participant watched each interaction partner make a choice between 'Yes' and 'No' to the question 'Would you like to deliver shocks to the other participant?'. As explained to the participant in the beginning of the experiment, the partner that chose 'Yes' would be informed about which image choice led to the delivery of shocks, making them the 'intentional' social interaction

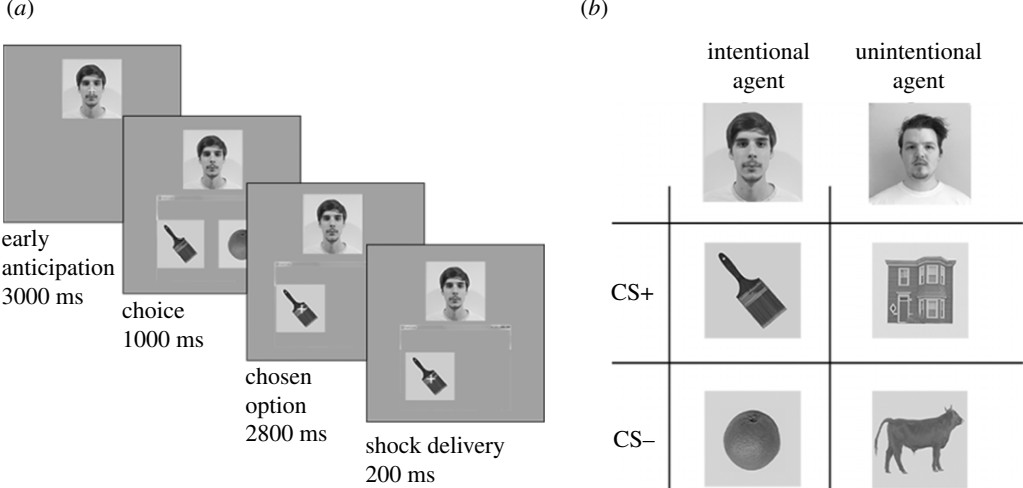

**Figure 2.** Experimental procedure. (*a*) On each trial, the subject passively observed an interaction partner (confederate) make a choice between two images. The confederate's face was always present on the screen and the window appeared as the confederate was allegedly making a decision. A fixation cross was present on the interaction partner's face in the early anticipation period and moved to the choice that was made during the choice period. This ensured that the participant viewed either the face or the choice during these periods, respectively. If the interaction partner chose an image that would be followed by the delivery of a shock, the shock was delivered for 200 ms and ended simultaneously with the choice image. (*b*) The 2 × 2 design.

partner. By contrast, the other interaction partner that chose 'No' (the unintentional interaction partner) had no information about the fact that they were delivering shocks, or which image would lead to the delivery of shocks. Participants could not continue with the experiment unless they responded correctly to a forced-choice question where they had to indicate whom (or none) of the interaction partners chose to deliver shocks. This question was repeated at the end of the threat learning phase, making sure every participant knew the identity of the intentional and unintentional interaction partners.

The experiment consisted of two phases: threat learning and extinction learning.

## 2.4. Threat learning

Threat learning consisted of 52 trials, 26 for each interaction partner. Each trial included three phases: (i) early anticipation: (3 s) presentation of a facial photograph of the interaction partner, (ii) choice: (1 s) presentation of the photograph of the interaction partner together with the two alternative options (CS+ and CS− stimuli) presented just below the face, and (iii) chosen option: (3 s) presentation of the photograph of the interaction partner and the choice (CS+ or the CS−) made by the interaction partner (see figure 2*a* for a more detailed overview).

For both interaction partners, choosing one of the images (CS+) (13 trials) caused the delivery of shocks to the participant 46% of the time (6 out of 13) during threat learning. Shocks were delivered at 2.8 s following the CS+ presentation. Note that in our previous work [1], shocks were delivered right before the presentation of the CS+ images. Choosing the other image (CS−, 13 trials) never caused the delivery of a shock to the participant. Inter-trial intervals (ITIs) were fixed to a length of 13 s, during which a fixation cross was presented. The onset of each trial was triggered by an fMRI pulse. We chose to use a fixed and relatively long ITI since in the trial-by-trial RSA method we are directly comparing two consecutive trials of each condition. When neural patterns are analysed on a single-trial basis, the results can be explained by the interference of temporal autocorrelations caused by temporal proximity. This ITI length has been shown to reduce intrinsic noise correlations substantially compared with event-related designs [18]. Additionally, we used a 'target' and 'filler' trial structure to ensure trials of each condition of interest were of equal distance to each other. Here, the trial order was fixed for each participant (counterbalanced across all participants) and consisted of a repeated sequence of seven target trials, with filler trials of the same stimuli in between [8,23]. The experiment thus consisted of alternating blocks of filler and target trials (electronic supplementary

material, S4). Target trials consisted of non-reinforced trials, meaning the participant did not receive any shocks. Only target trials were used in fMRI analyses to avoid the confounding effects of the delivery of shocks.

## 2.5. Extinction learning

The extinction learning task was identical to the threat learning task; however, the participants never received shocks. Participants were informed that the experiment would continue with an identical second phase after a break and were not informed about the absence of shocks. Hence, this phase served as an 'extinction learning' phase, in which the learned threat value was updated to safety.

## 2.6. Behavioural measures

Before participants were informed about which confederate was the 'intentional interaction partner', participants rated the likeability of confederate, on a scale between 1 (Not likeable at all) and 5 (Very likeable). The question was repeated after the threat learning phase, allowing us to assess changes in likeability after the threat interaction. After the scanning was completed, participants filled out a contingency awareness questionnaire to assess learning of associations between the CS+ images and shocks (electronic supplementary material, S3). Afterwards, each participant completed a questionnaire about their experiences through the threat learning phase including questions about the interaction partners, such as how angry the participant felt towards each of the interaction partners, how many shocks they would like to deliver to them if given the chance, and what the participant thought the motivation of the intentionally harming interaction partner was (electronic supplementary material, S2). At the end of the experiment, each participant was asked to rate how much they doubted that the experimental set-up was real (e.g. that the confederates really made choices), on a scale of 0% (never doubted) to 100% (always doubted).

## 2.7. Pupil dilation

During both phases, pupil dilation responses and eye-movements were recorded using an MR-compatible infrared eye tracker (SR Research), sampled at 250 Hz. The baseline pupil diameter was calculated as the average response over 500 ms preceding each trial. The response to each CS was calculated as the peak response during CS presentation (a window of 2.5 s) minus the baseline for that trial. Part of the data cleaning for pupil data consisted of replacing missing values (e.g. because of blinking, scanner artefacts). Data around blinks (100 ms before and after each segment of missing values) were deemed unreliable and together with the missing segments replaced by linear trend between the preceding and following data points. However, trials that ended up with substantial signal loss (more than 50% missing values after removing blink-related artefacts) were discarded entirely, and replaced using the linear trend between the preceding and following trial within that condition (to a maximum of 33% of trials per condition).

## 2.8. Functional MRI acquisition and pre-processing

A 3.0 T General Electrics MRI scanner and an 8-channel head-coil were used for scanning. Functional images were acquired using gradient echo-planar-imaging (EPI) (repetition time = 2000 ms, time echo = 28 ms, flip angle = 80°, 42 (estimated) sagittal slices with interleaved acquisition, $3.0 \times 3.0 \times 3.0$ mm) covering the whole brain. We performed higher order shimming. Five dummy scans were collected before each session. We used foam pads to minimize head motion. For anatomical scans, a high-dimensional T1-weighted image (repetition time = 6.4 ms, time to echo = 2.8 s, flip angle = 11°) was collected.

Functional MRI data were analysed using FSL (Oxford Centre for Functional MRI of the Brain (FMRIB) Software Library) software. Pre-processing was performed using FEAT (FMRI Expert Analysis Tool) version 5.0 and included slice-time correction, motion correction, high-pass filtering in the temporal domain (SIGMA = 100 s), and prewhitening. Structural images were transformed to MNI (Montreal Neurological Institute) standard space and co-registered to the functional images, using

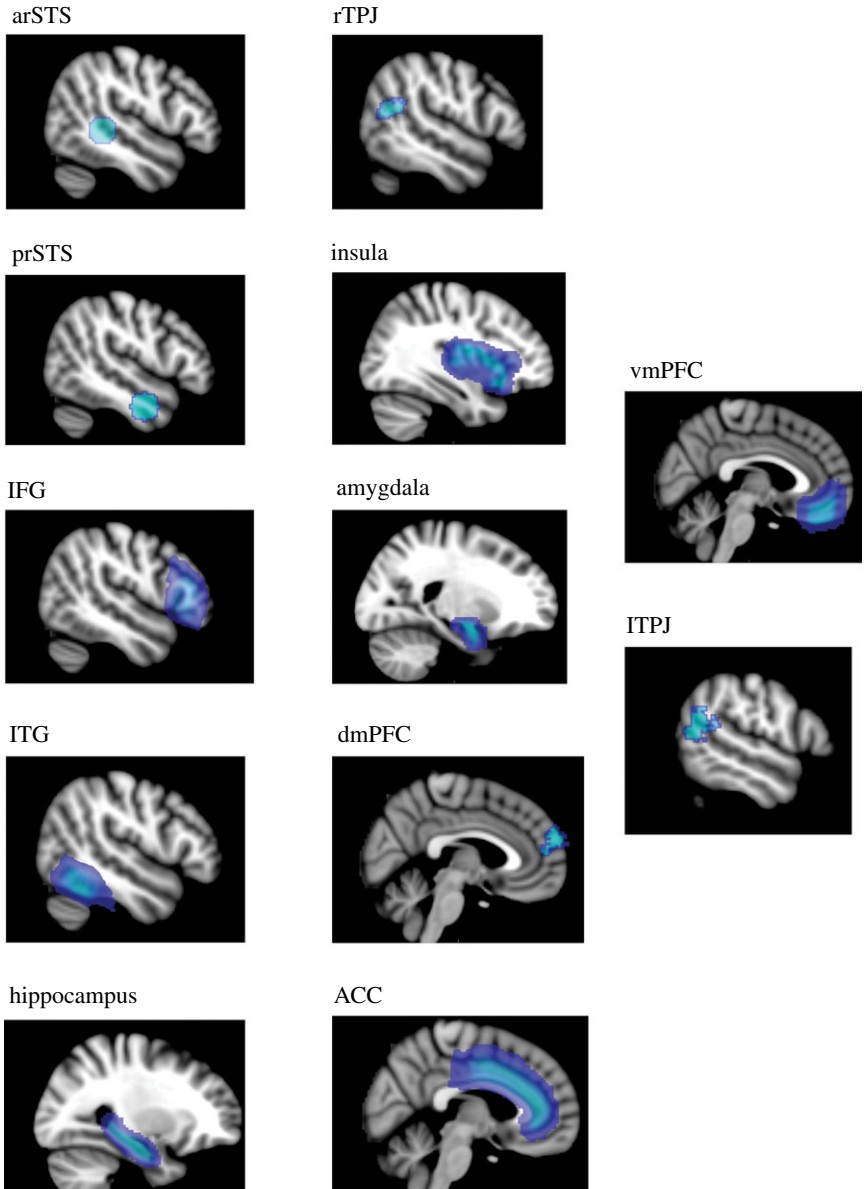

**Figure 3.** Visualization of regions of interest used as masks in the RSA, presented on the standard MNI brain used for the analyses.

FNIRT (FMRIB's Nonlinear Image Registration Tool, FSL). The resulting normalization parameters were applied to the functional images.

## 2.9. ROI selection

For threat learning regions, we chose the ACC, the amygdala, the insula, hippocampus and the vmPFC. For the regions related to intentionality processing, we chose left and right TPJ, vmPFC, dmPFC, anterior STS (arSTS) and posterior STS (prSTS) (figure 3). We included fusiform face area (FFA) only when analysing the early anticipation period (figure 2a) which consists of viewing the confederates' faces. We created anatomical ROIs using the Harvard–Oxford atlas and the regions unavailable in this tool (i.e. arSTS, and prSTS) were created 5 mm spherical masks based on coordinates of regions often cited in similar literature (acquired from the website Neurosynth, upon searching for the term 'intentions'). The coordinates were as follows:

arSTS = left -46 / -6 / -26 and right 50 / -4 / -34 combined
prSTS = left 22 / 44 / 36 and right 46 / -40 / -2 combined
Masks for all ROIs, apart from the left and right TPJ, were applied bilaterally.

## 2.10. Trial-by-trial representational similarity analysis

For both of the experimental phases (threat learning and extinction learning), we conducted separate voxel-wise whole-brain analyses. We modelled each trial as a separate regressor in a single generalized linear model (GLM). We created separate GLMs for the early anticipation and chosen option periods, modelling the blood-oxygen-level-dependent (BOLD) response starting on the onset and for the duration of the stimuli (3 s). GLMs included all trials of the experimental phase, in addition to the US (only for the threat learning phase) and motion parameters as nuisance regressors. The resulting parameter estimates for each trial were *z*-transformed and used to calculate pairwise Pearson correlations between spatial patterns of activation in each of the ROIs, for each trial of the experimental phase of interest. Overall, we had separate representational similarity matrices (RSMs) for each ROI, each subject, each experimental phase, and each period during a trial. We indexed the strength of pattern similarity as the strength of each of the correlations in the RSMs, for each experimental condition. As explained in the 'Threat learning' section above, RSMs were created only of target trials, which were not reinforced, and were presented in identical order in blocks. These target blocks were presented together with filler blocks that did not have a fixed order and included the reinforced trials. See the electronic supplementary material, S4 for more details.

## 2.11. Regression models for threat learning and extinction learning

In order to assess the contributions of intentionality, threat learning and their integration in different brain regions, we created template regression matrices to predict trial-by-trial similarity (figure 1*b*). Each matrix consisted of 28 × 28 elements, each element representing the similarity between the target trials (7 × 4, seven trials and four conditions). On the RSMs and regression templates, trials are grouped by condition for ease of visualization and modelling. Here, the diagonal line represents trial-by-trial similarity of each trial to itself ($i,j$) and is thus 1 for all RSMs (e.g. figure 1*a*). The off-diagonal line immediately adjacent to the diagonal line ($i,j+1$) represents the similarity between consecutive trials and was the trial-by-trial similarity measure we used in previous research [1]. Here, with the aid of the template regressors, we were able to model trials that are farther apart (e.g. $i,j+3$). See individual sections below, for more information on how each template regressor was created.

It is important to note that RSMs are diagonally mirrored (figure 1*a*), hence the correlation data between each trial is represented twice in a given RSM. Plus, each RSM contains a diagonal that represents a trial's similarity to itself (figure 1*a*, the diagonal line with perfect correlations). In this analysis, only the lower right triangle of the matrices, not including the diagonal, was used in the regression analyses.

### 2.11.1. Early anticipation period

We analysed the early anticipation and chosen option periods (figure 2*b*) of each trial separately to capture responses to the interaction partners' faces and action outcomes, respectively.

To predict the trial-by-trial neural correlations during the early anticipation period (figure 2*b*), in which the participant viewed a photograph of the interaction partner that was taking a turn, we created the following template regression matrices.

The first template regression matrix we used was of category information, modelling the similarity of a certain stimulus to itself, regardless of its experimental manipulation. This matrix was added as a regressor of control and tested for stimulus effects that are not related to the experimental manipulation.

1. Category information matrix, with the correlation 1 when element $i$ = element $j$. This regressor modelled the correlation between multiple presentations of the same stimulus.

The second matrix aimed to capture 'Intentional faces > Unintentional faces'.

2. Intentionality, with the correlation 1 when the intentional player's face was shown and, 0 elsewhere.

Here we aimed to detect the intentionality of the interaction partner, before they made a choice.

In the third matrix, the gradual integration of learning effects of intentionality were modelled, represented by an increase in correlations for the intentional confederate's face throughout the learning task (electronic supplementary material, S7, third panel on the top). Learning was defined as an increase in neural pattern correlations throughout the trials, excluding the first two habituation

trials where participants were not aware of the contingency between CS+ images and electrical shocks (see electronic supplementary material, S4 for a detailed breakdown of the threat learning phase).

3. Intentionality learning in time, with the correlation values for intentional face trials increasing from 0 to 1 throughout the experiment, and 0 elsewhere.

Here, we aimed to detect learning about the intentional social partner.

Lastly, we created a matrix that models learning effects on the unintentional social interaction partner (electronic supplementary material, S7, fourth panel on the top).

4. Unintentionality learning in time, with the correlation values for unintentional face trials increasing from 0 to 1 throughout the experiment, and 0 elsewhere.

Here, we aimed to detect learning about the unintentional social partner.

The matrices above were used in the analysis of the neuroimaging data for both the threat learning and extinction learning phases of the experiment.

### 2.11.2. Chosen option period

The template regression matrices common to both threat and extinction learning phases were as follows:

1. Category information matrix, with the correlation 1 when element $i =$ element $j$. This regressor modelled the correlation between multiple presentations of the same stimulus.

The second and third matrices aimed to capture 'Intentional choices > Unintentional choices' and 'CS+ > CS−' contrasts, respectively. These two templates model intentionality of choices (regardless of outcome), which aims to detect the intentional social interaction, and threat learning (regardless of intentionality), which aims to detect threat expectancy (figure 1$b$, 'Intentional > Unintentional' and 'CS+ > CS−').

2. Intentionality, with the correlation 1 when intentional choices were made and, 0 elsewhere.

The following template regression matrices were used for the threat learning phase:

3. CS, with the correlation 1 when a CS+ choice was made and, 0 elsewhere.

The following matrices were added to the regression model only for the threat learning phase:

In the fourth matrix, the gradual integration of learning effects of CS+ and intentionality were modelled similarly to the regressor used for the early anticipation period above. This regressor represented by an increase in correlations for CS+$_{intent}$ throughout the learning task, contrasted with the other stimuli (figure 1$b$, 'CS+$_{intentional}$ > CS$_{other}$'). Here, we aimed to detect a gradual learning effect, specifically for CS+$_{intentional}$.

4. Intentionality and CS+ integration in time, with the correlation values increasing from 0 to 1 through trials starting from trial 2 for intentional CS+ choices, and 0 elsewhere. In the first two trials of the threat learning phase, the participant only had information about the intentionality of choices but were naive to which of the CSs were paired with a shock (i.e. CS+). Therefore, the increase of correlations representing learning was modelled starting trial 2.

Lastly, we created a matrix that models learning effects on the unintentional CS+ (figure 1$b$, 'CS+$_{unintentional}$ > CS$_{other}$'). Here, we aimed to detect a gradual learning effect, specifically for CS+$_{unintentional}$.

5. Unintentionality and CS+ integration in time, with the correlation values increasing from 0 to 1 through trials starting from trials 2 for unintentional CS+ choices, and 0 elsewhere.

The following template regression matrices were added to the regression model only for the extinction learning phase (electronic supplementary material S10–12, top panel):

6. Intentionality and CS+ memory, with the correlation values decreasing from 1 to 0 through all CS+ intentional choice trials, and 0 elsewhere.

7.  Unintentional and CS+ memory, with the correlation values decreasing from 1 to 0 through all CS+ unintentional choice trials, and 0 elsewhere.

For each subject and each ROI, we regressed template regression matrices to RSMs, resulting in weights representing how much each of the template regression matrices was able to capture each RSM. To assess if we had any effects in the whole participant sample, we ran a one-sample $t$-test on these estimates to see if they significantly differed from zero across the sample.

All regression analyses described above were conducted using the package MASS [24] in R statistical language, using a linear model to assess the contribution of each regressor matrix. Each regressor matrix was Fisher-transformed. Since perfect correlation values return −Infinity or Infinity in such a transformation, we replaced these infinity values with 0.999 after the transformation. We used different models for each phase to predict the Fisher-transformed correlation values (see Trial-by-trial representational similarity analysis above) of each ROI using the formula

$$r_{(i,j)} = \beta_0 + \sum_{n=1}^{n_{max}} \beta_n \text{template}_{n(i,j)} + \epsilon_{(i,j)},$$

where $r$ is the estimated correlation, $i$ and $j$ trial number, and $\epsilon$ the error term. The significance of each coefficient in every model and ROI was tested by first calculating the $t$-statistics for each coefficient for each participant in each model and ROI. Second, for each model, ROI and coefficient, the distributions of the beta estimates from all participants were tested against zero using a one-sample $t$-test. Lastly, we used the Benjamini–Hochberg false discovery rate (FDR) correction [25] on the $p$-values acquired from each one-sample $t$-test to control for multiple comparisons within a model across ROIs (i.e. 60 tests in total per model), with the threshold of 0.05.

# 3. Results

## 3.1. Behavioural results

We used post-experimental questionnaires to measure participants' awareness of the relationship between images representing choices and their outcomes (i.e. contingency awareness, electronic supplementary material, S3), and evaluation of the intentional and unintentional interaction partners (electronic supplementary material, S2). We confirmed that the participants were aware of the relationship between the choice of each image and its outcome as they reported receiving a greater number of shocks from images paired with shocks (CS+ images), than ones with a safe outcome (CS− images) ($t_{23} = 3.17$, $p = 0.004$, $d = 0.65$). Next, we asked participants to report how much they expected to receive shocks and how many shocks they thought they received during the threat learning part of the experiment upon seeing each stimulus. Participants reported higher shock expectancy (CS [2] × Intentionality [2], $F_{1,24} = 8.74$, $p = 0.007$, $\eta^2 = 0.26$) and receiving more shocks from the intentional versus unintentional choices, irrespective of the outcome (figure 4$a$) (CS [2] × Intentionality [2], $F_{1,24} = 4.72$, $p = 0.047$, $\eta^2 = 0.16$), suggesting that the participants learned to associate intentional choices with a threat outcome more readily than unintentional ones.

To confirm whether the manipulation of intentionality was successful, participants were asked to answer questions about how they experienced the threat learning phase (electronic supplementary material, S3). The intentional shocks were rated more uncomfortable than unintentional ones (figure 4$b$) ($t_{23} = 3.02$, $p = 0.006$, $d = 0.62$). Participants were also angrier (figure 4$c$) ($t_{23} = 3.21$, $p = 0.004$, $d = 0.65$), and more revengeful (i.e. wanted to give back a greater number of shocks) (figure 4$d$) ($t_{23} = 3.33$, $p = 0.003$, $d = 0.68$) towards the intentional social partner. There was no significant difference between the decrease in likeability for the intentional and unintentional interaction partner after the threat learning paradigm. Lastly, we asked participants if they experienced any doubts about the information they were given about the experiment, specifically during the threat learning phase. Five participants out of the sample used for the analyses in this manuscript reported doubting more than 50% of the time they spent engaged in the social interaction (for the whole distribution see electronic supplementary material, S5a). When added to the above-reported statistical analyses, we found that the credibility did not affect any of the results reported above except for the estimated number of shocks received and experienced discomfort, which became non-significant (see electronic supplementary material, S5b). For additional reports of the open-ended post-experimental questions, see electronic supplementary material, S6.

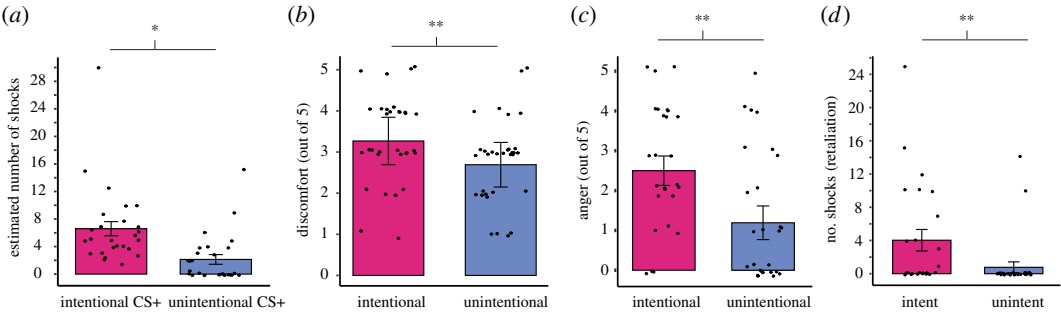

**Figure 4.** Behavioural results ($n = 26$). (*a*) Participants reported having received a greater number of shocks from intentional versus unintentional choices during the threat learning phase, (*b*) more discomfort from intentional than unintentional shocks, (*c*) feeling more anger towards intentional than unintentional interaction partner, (*d*) wanting to deliver a greater number of shocks to intentional than unintentional interaction partner, if given the chance. Error bars represent 95% CI, $^*p < 0.05$, $^{**}p < 0.01$.

## 3.2. Pupillometry results

We collected pupillometry data during the experiment to index learning, expecting to see greater arousal to CS+ images, indexed by greater pupil size (as per [1]). However, there were no significant differences between participant's mean pupil responses to the CS+ compared with the CS−, meaning that the learning effects we observed from the self-reported learning indexes were not reflected in physiological pupillary responses.

## 3.3. BOLD-fMRI results

Choices made by the interaction partners consisted of the following features: outcome (threat or safe) and intentionality (intentional or unintentional choice). Participants were informed about the intentionality of the decisions made before the experiment started; however, participants had to learn the outcome of choices themselves (i.e. the contingency between CS+ images and electrical shocks). This allowed us to formulate specific predictions regarding the timeline of the integration of our two features of interest: social information and learned threat value. We formulated the regression matrices to predict an increase in trial-by-trial pattern correlations to increase in response to certain stimuli but not others. The regressor intentional > unintentional (figure 1*b*) predicts trial-by-trial similarity to be higher for intentional choices regardless of threat value, than unintentional ones. The regressor CS+ > CS− predicts trial-by-trial similarity to be higher for threat choices, regardless of intentionality, than safe ones. The regressor CS+$_{intentional}$ > CS$_{other}$ predicts trial-by-trial similarity to be higher for intentional threat choices, and to increase over time. The regressor CS+$_{unintentional}$ > CS$_{other}$ predicts trial-by-trial similarity to be higher for unintentional threat choices, and to increase over time. To compare the specificity of each brain region's involvement in these predictions, we regressed RSA matrices (representational similarity matrix, or RSM) onto regressor matrices representing our predictions (template regression matrices; figure 1*b*) to capture the action outcome and intentionality, as well as their integration. We created RSMs from activity during 'early anticipation' and 'chosen option' periods (figure 2*a*) within a trial, separately (see Material and methods).

## 3.4. RSA regression: threat learning phase

### 3.4.1. Early anticipation period

For the early anticipation period (figure 2*a*), during which participants viewed photographs of social interaction partners before a choice was made, we found a significant effect of templates for category information, learning of intentionality and unintentionality (i.e. gradual increase in correlation) in the FFA ($t_{21} = -5.09$, $p < 0.001$, $d = 1.11$) and inferior temporal gyrus (ITG) ($t_{21} = 4.96$, $p < 0.001$, $d = 1.08$) that survived an FDR correction (electronic supplementary material, S7). We found no significant differences between intentional and unintentional learning in either of these regions. No significant variance was explained by any of the other regressor templates, for any other ROI. Thus, we captured an increase in trial-by-trial correlations in response to both intentional and unintentional faces in the FFA and our control region ITG.

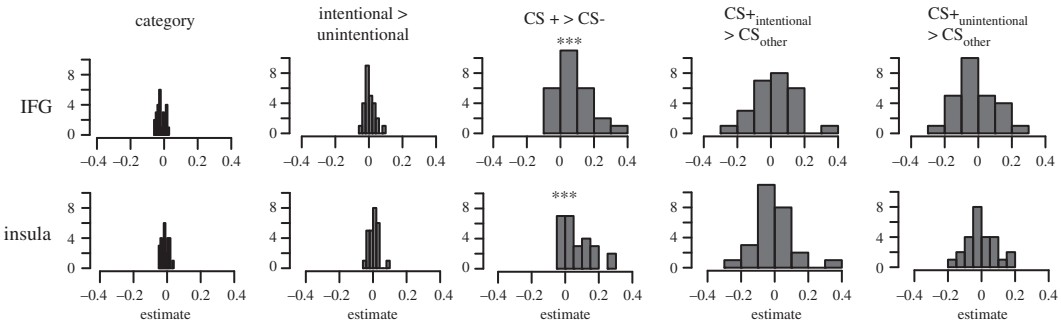

**Figure 5.** RSM's and regression results ($n = 26$). Distribution of regression estimates for all participants, for each template regression matrix, in the IFG and insula. $^{***}p < 0.001$, FDR-corrected.

### 3.4.2. Chosen option period

Unexpectedly, we saw no effects of intentionality when analysing the chosen option period. Only the CS type and category regressors survived an FDR correction, and explained significant variance in the trial-by-trial activation patterns (electronic supplementary material, S8). Category information was represented in the IFG ($t_{25} = -4.33$, $p < 0.001$, $d = 0.85$), whereas the insula ($t_{25} = 4.02$, $p < 0.001$, $d = 0.78$) and the IFG ($t_{25} = 4.31$, $p < 0.001$, $d = 0.84$) (figure 5) showed higher neural pattern similarity for CS+ in contrast to CS− trials.

In addition to the template regression, and in order to be able to provide comparisons with previous research [1], we also performed regular repeated-measures ANOVAs on the correlations between consecutive trials (1–2, 2–3, etc.) ($i,j + 1$) of a stimulus (figure 1a). Again, we detected no significant effects of intentionality. We observed greater trial-by-trial correlation in the ACC ($F_{1,25} = 5.55$, $p = 0.03$, $\eta^2 = 0.18$), the insula ($F_{1,25} = 17.04$, $p < 0.001$, $\eta^2 = 0.40$) (figure 1a), the arSTS ($F_{1,25} = 4.04$, $p = 0.05$, $\eta^2 = 0.14$), and the vmPFC ($F_{1,25} = 5.72$, $p = 0.02$, $\eta^2 = 0.19$). Out of these, only the effect in the insula survived the FDR correction. Despite being weaker, our results are consistent with previous findings showing neural pattern formation in response to threat learning, where an increase in correlations was observed as a main effect of CS type in the ACC, the insula [1] and the vmPFC [8] (for a direct comparison between [1] and the current article, see electronic supplementary material, S13). We failed to replicate our previous findings which showed an increase of neural pattern correlations specific to intentional threat choices in the IFG, the insula, the dmPFC and the arSTS.

We used a univariate GLM approach to explore intentionality-related activity in the brain regions that were not included in our initial set of ROIs. We modelled the target trials, the filler trials, the anticipation period, the chosen option period, and US delivery. For a more detailed account of the results and the methods of the univariate analyses, see electronic supplementary material, S14–S16. Although, we did not find a main effect of intentionality (Intentional > Unintentional), the CS+$_{intentional}$ > CS+$_{unintentional}$, engaged the insula, the dmPFC and the right temporoparietal junction (rTPJ) (see electronic supplementary material, S14 and S15). The effects in the insula were also present for the CS+ > CS− (i.e. CS+ activity regardless of intentionality) (electronic supplementary material, S14 and S16). The contrast CS+$_{unintentional}$ > CS+$_{intentional}$ yielded no significant clusters of activity.

## 3.5. RSA regression: extinction learning phase

### 3.5.1. Early anticipation period

For the anticipation period (i.e. the viewing of a face, before a choice was made) in the extinction learning phase, we chose to test only for an increase in correlations (e.g. reflecting extinction learning), rather than a decrease (e.g. reflecting a decline in threat), because neural correlation patterns in this anticipation phase failed to show evidence of threat learning (i.e. an increase in pattern similarity) for the intentionality manipulation. A decrease in neural pattern correlations during the extinction learning phase would therefore be hard to interpret. We observed an effect of category information ($t_{21} = 5.98$, $p < 0.001$, $d = 1.89$) and unintentional learning ($t_{21} = -5.48$, $p < 0.001$, $d = 1.73$) in the ITG. The template modelling a gradual increase in pattern correlations in response to the intentional faces detected activity in the hippocampus ($t_{21} = -5.13$, $p < 0.001$, $d = 1.62$). This finding indicates a novel memory trace regarding safety about the intentional confederate in the hippocampus, upon an interaction in which no shocks were delivered.

### 3.5.2. Chosen option period

We examined neural pattern similarity during the chosen option period in the extinction learning phase based on three different hypotheses, for which we created three different sets of regressors (electronic supplementary material, S10–12, top panel): (i) 'extinction of threat value', where the neural correlations would decrease over time indicating decline in threat value; (ii) 'resistance to extinction learning', where neural pattern correlations for CS+$_{intent}$ choices would stay stable while that of other CSs decrease over time, and (iii) 'extinction learning', where neural pattern correlations increase over time, representing novel learning of safety. For the extinction of threat value, the observed differential decrease in neural pattern similarity over time showed significant effects in the amygdala ($t_{25} = 4.16$, $p < 0.001$, $d = 0.81$), and the ACC ($t_{25} = -4.60$, $p < 0.001$, $d = 0.90$) for the CS+$_{intent}$. We found a decrease in correlations in response to the CS+$_{unintent}$ in the ITG ($t_{25} = 4.73$, $p < 0.001$, $d = 0.92$). In the insula, the rTPJ and the IFG, a decrease of trial-by-trial correlations was observed for both intentional (IFG: $t_{25} = 4.91$, $p < 0.001$, $d = 0.96$; rTPJ: $t_{25} = 4.35$, $p < 0.001$, $d = 0.85$; insula: $t_{25} = 4.92$, $p < 0.001$, $d = 0.99$) and unintentional CS+s (IFG: $t_{25} = 4.90$, $p < 0.001$, $d = 0.96$; rTPJ: $t_{25} = 3.59$, $p < 0.001$, $d = 0.70$; insula: $t_{25} = 4.05$, $p < 0.001$, $d = 0.79$). Category information was captured in the IFG ($t_{25} = -5.35$, $p < 0.001$, $d = 1.05$), the insula ($t_{25} = -5.04$, $p < 0.001$, $d = 0.98$), the ITG ($t_{25} = -4.79$, $p < 0.001$, $d = 0.94$), the arSTS ($t_{25} = -3.36$, $p = 0.002$, $d = 0.66$), the prSTS ($t_{25} = -4.74$, $p < 0.001$, $d = 0.93$) and the lTPJ ($t_{25} = -3.71$, $p = 0.001$, $d = 0.72$). None of the other significant regressors (electronic supplementary material, S10) survived FDR correction. Testing the resistance to extinction learning, none of the significant findings survived the FDR correction (electronic supplementary material, S11), except for the category effect in the prSTS ($t_{25} = -4.51$, $p < 0.001$, $d = 0.88$).

## 4. Discussion

We designed a social interaction task in which participants learned the threat value of stimuli through experiencing the outcomes of another individual's actions, where one partner caused the outcomes intentionally, and the other unintentionally. We showed that participants expected and reported a higher frequency of intentional compared with unintentional harmful choices, and after the social interaction was complete, participants reported more anger and desire for revenge towards the intentional partner. These findings showed that threat learning through an intentional action changed how the social partner and their actions were explicitly evaluated. Using a trial-by-trial application of RSA, we observed a differentiation between threatening and safe choices in pattern correlations over time in the insula and the IFG, replicating previous findings implicating a role for the insula and IFG in threat learning [1,16].

In the current study, we collected both pupil responses, and explicit measures (i.e. a contingency awareness questionnaire, electronic supplementary material, S3) to index learning. We found no differences in pupil responses to the CS+ as compared with CS−, showing that participants on average failed to express this implicit marker of learning. We did, however, find that participants successfully reported the relationship between the CS+ images and the delivery of electrical shocks. Previous studies have argued that such explicit awareness is important to capture differences in physiological responses to learned stimuli [26], leading some researchers to exclude participants failing to show learning effects based on explicit awareness [11]. Research has shown that pupil responses can successfully predict trial-by-trial neural pattern formation during threat learning [8]. Interestingly, in the current study, we showed a selective increase in neural pattern similarity to the threat stimulus even in the absence of such pupil responses, suggesting that neural similarity measures may be more sensitive in detecting subtle learning effects than peripheral (e.g. pupil dilation) read-outs of threat.

Our findings suggest that the IFG might represent the threat value of actions through an increase in trial-by-trial neural pattern correlations (figure 5). This suggestion is consistent with previous work using single-trial fMRI data during threat learning, reporting discriminative voxel pattern formation to CS+ stimuli compared with the CS− in the IFG [16,27]. Apart from its involvement in threat processing, IFG activation has been reported in studies involving language [28], sensory processing [29] and social cognition [10]. These functions align well with the observation that this region is involved in understanding others' mind states and actions, as well as communicating one's own.

Here, and in our previous study [1], the RSA findings are reported for trial blocks that did not include shocks even if a threat choice (i.e. CS+) was made (see Material and methods sections for 'target' and 'filler' trials, as well as electronic supplementary material, S4). Thus, the intentionality effects in our

previous work were responses for unreinforced CS+ choices that were associated with a threat outcome when chosen at other instances but did not actually deliver shocks. In other words, given that learning took place, participants were aware that a threat choice was made, but received no shock. An important difference between the previous study and the current one was the timing of the shocks. In the current study, shocks were delivered *after* the threat choice was presented, whereas in our previous study shocks were delivered right before. Thus, in the current study participants learned that the presentation of the threat choice presentation *led* to a shock, whereas in the previous study the participants learned that the threat choice presentation was *preceded* by a shock. Hence, when we report our findings for the threat choice presentation period, the process we capture in the two studies are in fact different. In comparison with our previous study, here we captured the *expectation* of a threatening outcome, rather than the *realization* that outcome was made and no aversive stimulus was delivered. The fact that the IFG represented intentionality when the participants realize a threat choice was made, although there was no aversive stimuli, might indicate IFG's involvement in relief of not experiencing the aversive outcome from an intentional action. The lack of intentionality effects in the IFG in the current study thus hints towards what kind of information might be represented in the IFG. If there is an expectation of an aversive outcome, the IFG develops a representation of threat value (indexed by neural pattern similarity increase) regardless of the intentionality behind it (figure 5). If there is a realization of lack of aversive outcome from an intentional threatening event, the IFG integrates intentionality with the threat value [1]. Supporting this possibility, a recent study comparing social and physical pain suggests that relief from physical, but not social, pain engages the IFG [30]. Unfortunately, our data do not allow us to further investigate this possibility.

In addition to the IFG, we show that learning about threat choice outcomes (i.e. CS+), compared with safe ones (i.e. CS−), leads to an increase in neural pattern correlations in the insula (figures 1a and 5). This finding is in line with previous research that implicates the insula's role in both direct and social fear learning [31]. Interestingly, we found no difference in pupil responses to threat as compared with safe stimuli, but we did show neural representation of the CS+ in the insula.

Our replication efforts point to several important considerations that should be noted when implementing paradigms similar to ours in future studies. Firstly, future work should examine potential effects of the timing of aversive stimulation by manipulating the onset of the shock in relation to the CS+. Secondly, stopping rules based on learning should be considered when recruiting participants. As we failed to replicate learning effects in pupillometry data, this should be considered as a potential factor in driving the difference between the results of the two studies. Finally, considering the well-documented effects of intentionality on pain and the involvement of the insula [13,32], future studies should consider using painful rather than uncomfortable aversive stimuli.

In the current study, we aimed to replicate and extend a previous study [1] by applying a regression-based RSA to a socio-interactive threat learning paradigm. We showed that the trial-by-trial RSA method in conjunction with the model-based analysis has the potential to disentangle brain mechanisms underlying the many facets involved in threat learning in a social context. Although we replicated the behavioural effects of intentionally [1], we did not observe the expected neural effects, which might be related to a lack of power and/or methodological differences as compared with previous work. Unlike the CS+, the intentionality manipulation was not accompanied by a reminder cue (e.g. reminding the participant about the intentionality of the current trial). The lack of a reminder might have contributed to a weaker effect, especially in comparison with the conditioning effect. Indeed, to capture the effects of intentionality in the presence of a salient stimulus like the CS+, a greater sample size might be necessary. Our previous study [1], where we were able to capture the effects of intentionality in fear learning, contained a greater sample size. Future research could include reminders about the intentionality and/or a larger sample size. Yet, as predicted, our methodological approach captured an increase in trial-by-trial neural pattern correlations to learned threat. Taken together, our study extends and bridges previous research on threat learning and social interactions, and provides new insights into the processes affecting emotional learning during social interactions.

Data accessibility. Data and analysis scripts are available from the Dryad Digital Repository: https://doi.org/10.5061/dryad. k6djh9w58 [33].

Authors' contributions. I.U., R.M.V. and A.O. designed the experiments. I.U. collected the data and drafted the manuscript. I.U., R.M.V., L.d.B. and N.B. performed the data analysis. All authors provided revisions and approved the final version of the manuscript for submission.

Competing interests. The authors declare no competing interests.

Funding. This research was supported by the Knut and Alice Wallenberg Foundation (KAW grant no. 2014.0237), a European Research Council Starting Grant (284366; Emotional Learning in Social Interaction project), and a Consolidator Grant (2018-00877) from Swedish Research Foundation (VR) to A.O. R.M.V. was supported by

a Netherlands Organization for Scientific Research (NWO Veni) grant (016.veni.195.246) and A.G. was supported by a project grant (K822206083) from the Swedish Research Council. N.B. was supported by the Swedish Research Council (Vetenskapsrådet; 2017-06146). L.d.B. was supported by the Swedish Research Council (Vetenskapsrådet; 521-2013-2589).

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
