## [Peer Review File · Royal Society Open Science]

Review History

RSOS-202116.R0 (Original submission)

Review form: Reviewer 1

Is the manuscript scientifically sound in its present form?

Yes

Are the interpretations and conclusions justified by the results?

Yes

Is the language acceptable?

Yes

Do you have any ethical concerns with this paper?

No

Have you any concerns about statistical analyses in this paper?

No

Recommendation?

Reject

Comments to the Author(s)

Summary:

In the current study, the authors explored how people perceive and learn about threatening actions of others. They explored the behavioral, physiological, and neural responses to threat learning by watching confederates make decisions which intentionally or unintentionally resulted in a shock to participants. Importantly, this study attempted to directly replicate an earlier study (published in *Cerebral Cortex*) from the same group. The only difference is that a different sample of participants tested, and additional analyses were done in order to reveal the underlying aspects of the social interactions in the experiment. Although the behavioral results supported their hypotheses and replicated their earlier findings, the pupillometry and neuroimaging results did not match up as well. It was interesting to see the pupillometry portion of the study, and the representational similarity analyses was very thoroughly implemented. However, there are a number of critiques that need to be addressed.

I didn't realize that this was a near-exact replication until I looked at the author's previous study published in *Cerebral Cortex* and saw the task and stimuli. This is not a bad thing at all! However readers will want to know how similar this is to previously-published study. The authors should be more explicit about this, in the abstract, the methods, and discussion.

Also, the RSA technique is said to be novel but it probably is not.

Critiques:

Methodological

1. I would like a citation for why these ToM ROIs that were selected. STS seems like an odd choice, given its strong role in basic language processing. Why not the ATL instead? It has been linked to person knowledge and ToM for over 20 years.
 2. Are the two vmPFC ROIs the same for the threat learning and ToM analyses? A visual depiction of the ROIs would be helpful.
 3. It is stated several times that this regression-based RSA is novel. How does it differ from previous uses (Tamir et al., 2016; Brooks & Freeman, 2018; Stolier & Freeman, 2016; Lee Masson et al., 2018; Bracci et al., 2015)? See Popal et al., 2019 for a review of RSA studies that used multiple regression. If, as I suspect, the method is not novel, please alter the text to reflect this.
 4. It is good that the ITI is long, but if the trials were not randomized couldn't there still be an increase in false positives, especially if within-run comparisons were made (Mumford et al., 2014)? Were multiple runs acquired and were trial-by-trial comparisons done between runs to somewhat alleviate this issue?
 5. I would be useful to provide visual depiction of the ROIs to understand exactly what areas you chose as your ROIs (most researchers aren't very good at visualizing MNI coordinates).
- Discussion/interpretation
6. The IFG finding needs to be unraveled. The left IFG has been linked to language production for over 100 years. Please add text to the discussion. Note that the authors state that the IFG finding replicates Visser's study but I checked and Visser does not mention the IFG in their study.
 7. ToM is usually right lateralized (Corbetta et al., 2008; Boccadoro et al., 2019; Scholz et al., 2009). Laterality wasn't discussed.
 8. The discussion is weak. It should discuss the current findings in greater detail. The current version is very much oriented towards discussing how the findings are similar or different from this group's prior study. I like this and it would actually help readers if this were to be put into table format. As a reader, I would like to know: should I use this task and if I do,

what can I expect to find? However, you should also elaborate on the current findings for readers who are unaware of your Cerebral Cortex study.

Minor:

- I was confused why the authors called extinction learning “safety learning”. I have never heard of that term used before, whereas extinction learning is widely used. It was unclear why this distinction was made.
- When describe the scan acquisition parameters, the authors had a typo and I assume mixed TR with total scan time for the T1.
- The “BOLD-fMRI results” section seems to just be methods in how the RSMs are created and how RSA will be done. Perhaps it would make the paper and the RSA easier to understand if you moved this to the methods section, with a subsection for RSA and RSM creation. Maybe a sub-subsection for each RSM?
- Figure 2 is hard to read. I recommend (1) decreasing the number of increments on the y-axis for (a) and (d)...it’s crowded and hard to read ; (2) increase the resolution (it isn’t very crisp); and (3) if allowed, use color to make it prettier.
- Figure 3 is too low resolution. It will look better in print if efforts are made to improve this.
- It would be better to provide the supplementary material document in a more accessible format, such as .doc so that people without Macs can open it. I was unable to open it on my PC.

Review form: Reviewer 2

Is the manuscript scientifically sound in its present form?

Yes

Are the interpretations and conclusions justified by the results?

No

Is the language acceptable?

Yes

Do you have any ethical concerns with this paper?

No

Have you any concerns about statistical analyses in this paper?

Yes

Recommendation?

Major revision is needed (please make suggestions in comments)

Comments to the Author(s)

This paper attempts to replicate and extend a previous study using a threat learning paradigm where there are two confederates, one that can provide shocks (UCS) with intent, and another one, whose UCS is not provided with intent. Data is explored with RSM and a novel RSA regression method. Perhaps it is somewhat because of the tricky analyses, but I find myself with many questions and concerns when reading this paper. Foremost about the analyses and how to interpret the results. I hope by addressing these concerns, the authors will be helped in improving the readability of the paper.

Methods:

P9 L162: “As an independent index of threat learning we collected pupillometry data, excluding participants who had more than 33% of trials of any condition missing (missing trial defined as > 50% missing sample for that trial) (n =3)”

At this point in the paper, the “conditions” are not defined. Which are they? CS+ (intent), CS- (intent), CS+ (no intent), CS- (intent)?

I guess the same two individuals acted as confederates for all participants. However, this is unclear from the ms.

P9 L174: “These categories were chosen as they are represented in different regions of the brain [20], and ensuring there are no generalization effects between the CS’s during learning.” Ensuring seems bold. Perhaps: “...in different regions of the brain, in order to minimize generalization effects..” What generalization effects are you trying to minimize?

P11: I am confused by the “target” and “filler” structure. It is used to “ensure trials of each condition of interest were of equal distance to each other”. Do you mean in order to make the same trial not appear many times in a row?

Target trials are non-reinforced trials. However, filler trials are not all reinforced. So I guess that some filler trials are omitted and classified as filler trials in order to make an equal amount of trials for the different conditions. Is this true? If so, please state in the ms. Supp mat 4 also confuses me as it depicts block of trials. Is a block of target trials 7 trials long? Does Sup mat 4 depict the actual order of blocks? In that case the experiment actually starts with a long habituation period during which no shocks are delivered. How many trials are in a filler block? Threat learning should have $13 * 4 = 52$ trials. How are these divided on the different blocks? I may be missing something obvious, but I am confused about this.

P13 L263: “...discarded and using the linear point...” What does this mean? Which point?

P14 L283: Please define “ToM network”.

P15: Regression models.

I have trouble understanding this. I guess you are trying to predict the similarity from trial to trial within your ROI:s by a regression matrices representing the different trials ordered into your regressions in order to represent the different effects that you are interested in. However, the description confuses me.

“...we created template regression matrices for each feature of interest (figure 3 b) with elements i and j , each representing a trial in the experiment [23], where the correlation between the same stimuli would be represented at $i = j$ indices of the matrix. Each matrix thus consisted of $24 * 24$ elements (one for each target trial of each condition).

Please specify “feature of interest”, or state that you will have different for each period of your trials.

Don’t you mean that $i=j$ indices represent the correlation between the same stimuli at the same trial? Indices i and j represent trials. So $(i,j+1)$ will represent the correlation between consecutive trials for the same stimuli, the way they are ordered in fig 3b.

“Each matrix thus consisted of $24 * 24$ elements (one for each target trial of each condition).” Please define “target trial”.

P16 EARLY ANTICIPATION PERIOD.

Here are 4 regressors created, some which are not depicted in fig 3b. These are the “feature of interest” for the early anticipation period?

About the “intentionality increase in time” and “Unintentionality increase in time”. Intentionality is learned at the start of the experiment. Thus, neither intentionality or unintentionality can increase over time. However, the participants feelings or responses to the faces of the confederate can of course change over the experiment, but these regressors seem badly named. Please explain.

L 335 “Note that we did not conduct the analysis using a “decrease” of correlational values for faces for the extinction phase, since we saw no pattern formation during learning.” What pattern formation are you referring to here?

L 337 “Instead, we chose to only report the regressors for increase of correlation, which represent a novel representation of safety during extinction.” How can we be sure of that?

P17: CHOSEN OPTION PERIOD

Again I am confused by the regressors. Why not include a CS+ correlation increase over trials, which in my mind might represent the development of a conditioned response? Instead the change over trials are surmised to interact with intentionality. Also, as before, I don’t see how intentionality or unintentionality can develop over time as it is learned from the outset.

P18: L360- About extinction. Is regressor 6 and 7 added to the ones before or do they replace regressor 4 and 5 on page 17?

Results:

L 394: “Participants reported higher shock expectancy ($F(1,24) = 8.74, p = 0.007, \eta^2 = 0.26$) and receiving more shocks from the intentional vs. unintentional choices, irrespective of the outcome (figure 2 a)”

Why is this an ANOVA? Are both CS+ and CS- lumped together? What not a t-test between CS+ (intent) and CS+ (no intent)?

L 398: “Overall, participants thought CS+’s were chosen more often, regardless of intentionality.” Statistics?

L 409: “Five participants reported doubting more than 50% of the time they spent engaged in the social interaction (for the whole distribution see electronic supplementary material 5 a). When added to the above reported statistical analyses, we found that the credibility did not affect any of the results reported above except for...”

So five participants were removed from the behavioral results. This is not reported under participants in the methods section. Were they removed in any other analyses?

L 428-429 The pupillometry results were insignificant, but $p < 0.001$ in reference [1]. Do you have any ideas why? It is a big difference. From the psychophysiological data it appears you have achieved contingency awareness, but not fear conditioning.

L434: “This allowed us to formulate specific predictions regarding the timeline of the integration of our two features of interest: social information and learned threat value.”

Please specify these prediction. Do you mean the formulation of regressor matrices?

L480 Please define the abbreviation "FFA".

L 489-498 One of the reasons for this study is "...to replicate these findings in a novel dataset...". Here is where this is attempted. The authors conclude that "Despite being weaker, our results are consistent with previous findings of neural pattern formation in response to threat learning, where an increase in correlations were observed as a main effect of CS type in the ACC, the insula [1] and the vmPFC." However, in [1] the CS+ (intent) appeared to differ from the other conditions. In the present study, according to the one graph that is shown in figure (3a), it appears both CS+ differ from both CS-. Does this possibly change the authors conclusion of replication? Does it look the same in the ACC and vmPFC that did not survive FDR correction. Oh wait, this is what is mentioned in the discussion on L 540? "We could not, however, replicate the increase in neural pattern correlations specific to the intentional threatening actions." Perhaps this could be mentioned also here?

Around here, please refer to figure 3a. As it stands, it is not referred to in the text.

L500- "For the anticipation period (i.e., the viewing of a face, before a choice was made) in the extinction phase we chose to test only for an increase in correlations (e.g., reflecting safety learning), rather than a decrease (e.g., reflecting a decline in threat), because neural correlation patterns in this anticipation phase failed to show evidence of threat learning (i.e., an increase in pattern similarity) in the first place."

Where is this failure to show threat learning reported?

Discussion:

I have a hard time evaluating the conclusions with all my questions on the data analysis unanswered. It appears the RSM and regression results are not discussed?

Decision letter (RSOS-202116.R0)

Dear Ms Undeger

The Editors assigned to your paper RSOS-202116 "Model-based representational similarity analysis of BOLD fMRI captures threat learning in social interactions" have now received comments from reviewers and would like you to revise the paper in accordance with the reviewer comments and any comments from the Editors. Please note this decision does not guarantee eventual acceptance.

Please submit your revised manuscript and required files (see below) no later than 21 days from today's (ie 23-Feb-2021) date. Note: the ScholarOne system will 'lock' if submission of the revision is attempted 21 or more days after the deadline. If you do not think you will be able to meet this deadline please contact the editorial office immediately.

on behalf of Dr Jonathan Roiser (Associate Editor) and Essi Viding (Subject Editor)
openscience@royalsociety.org

Associate Editor Comments to Author (Dr Jonathan Roiser):

Associate Editor: 1

Comments to the Author:

Thank you for submitting your work to RSOS. Your manuscript has been reviewed by two experts in the field, who felt that the work was of interest but also raised several concerns and queries. We would be willing to consider a revised version that addresses these points, but at this stage cannot guarantee that the manuscript will be accepted, as some of the comments are quite critical. Please ensure that all reviewer queries are addressed in a point-by-point fashion in your revised submission.

Reviewer comments to Author:

Reviewer: 1

Comments to the Author(s)

Summary:

In the current study, the authors explored how people perceive and learn about threatening actions of others. They explored the behavioral, physiological, and neural responses to threat learning by watching confederates make decisions which intentionally or unintentionally resulted in a shock to participants. Importantly, this study attempted to directly replicate an earlier study (published in Cerebral Cortex) from the same group. The only difference is that a different sample of participants tested, and additional analyses were done in order to reveal the underlying aspects of the social interactions in the experiment. Although the behavioral results supported their hypotheses and replicated their earlier findings, the pupillometry and neuroimaging results did not match up as well. It was interesting to see the pupillometry portion

of the study, and the representational similarity analyses was very thoroughly implemented. However, there are a number of critiques that need to be addressed.

I didn't realize that this was a near-exact replication until I looked at the author's previous study published in *Cerebral Cortex* and saw the task and stimuli. This is not a bad thing at all! However readers will want to know how similar this is to previously-published study. The authors should be more explicit about this, in the abstract, the methods, and discussion.

Also, the RSA technique is said to be novel but it probably is not.

Critiques:

Methodological

1. I would like a citation for why these ToM ROIs that were selected. STS seems like an odd choice, given its strong role in basic language processing. Why not the ATL instead? It has been linked to person knowledge and ToM for over 20 years.
2. Are the two vmPFC ROIs the same for the threat learning and ToM analyses? A visual depiction of the ROIs would be helpful.
3. It is stated several times that this regression-based RSA is novel. How does it differ from previous uses (Tamir et al., 2016; Brooks & Freeman, 2018; Stolier & Freeman, 2016; Lee Masson et al., 2018; Bracci et al., 2015)? See Popal et al., 2019 for a review of RSA studies that used multiple regression. If, as I suspect, the method is not novel, please alter the text to reflect this.
4. It is good that the ITI is long, but if the trials were not randomized couldn't there still be an increase in false positives, especially if within-run comparisons were made (Mumford et al., 2014)? Were multiple runs acquired and were trial-by-trial comparisons done between runs to somewhat alleviate this issue?
5. I would be useful to provide visual depiction of the ROIs to understand exactly what areas you chose as your ROIs (most researchers aren't very good at visualizing MNI coordinates).

Discussion/interpretation

6. The IFG finding needs to be unraveled. The left IFG has been linked to language production for over 100 years. Please add text to the discussion. Note that the authors state that the IFG finding replicates Visser's study but I checked and Visser does not mention the IFG in their study.
7. ToM is usually right lateralized (Corbetta et al., 2008; Boccadoro et al., 2019; Scholz et al., 2009). Laterality wasn't discussed.
8. The discussion is weak. It should discuss the current findings in greater detail. The current version is very much oriented towards discussing how the findings are similar or different from this group's prior study. I like this and it would actually help readers if this were to be put into table format. As a reader, I would like to know: should I use this task and if I do, what can I expect to find? However, you should also elaborate on the current findings for readers who are unaware of your *Cerebral Cortex* study.

Minor:

- I was confused why the authors called extinction learning "safety learning". I have never heard of that term used before, whereas extinction learning is widely used. It was unclear why this distinction was made.
- When describe the scan acquisition parameters, the authors had a typo and I assume mixed TR with total scan time for the T1.
- The "BOLD-fMRI results" section seems to just be methods in how the RSMs are created and how RSA will be done. Perhaps it would make the paper and the RSA easier to understand if you moved this to the methods section, with a subsection for RSA and RSM creation. Maybe a sub-subsection for each RSM?
- Figure 2 is hard to read. I recommend (1) decreasing the number of increments on the y-axis for (a) and (d)...it's crowded and hard to read ; (2) increase the resolution (it isn't very crisp); and (3) if allowed, use color to make it prettier.

- Figure 3 is too low resolution. It will look better in print if efforts are made to improve this.
- It would be better to provide the supplementary material document in a more accessible format, such as .doc so that people without Macs can open it. I was unable to open it on my PC.

Reviewer: 2

Comments to the Author(s)

This paper attempts to replicate and extend a previous study using a threat learning paradigm where there are two confederates, one that can provide shocks (UCS) with intent, and another one, whose UCS is not provided with intent. Data is explored with RSM and a novel RSA regression method. Perhaps it is somewhat because of the tricky analyses, but I find myself with many questions and concerns when reading this paper. Foremost about the analyses and how to interpret the results. I hope by addressing these concerns, the authors will be helped in improving the readability of the paper.

Methods:

P9 L162: "As an independent index of threat learning we collected pupillometry data, excluding participants who had more than 33% of trials of any condition missing (missing trial defined as > 50% missing sample for that trial) (n =3)"

At this point in the paper, the "conditions" are not defined. Which are they? CS+ (intent), CS- (intent), CS+ (no intent), CS- (intent)?

I guess the same two individuals acted as confederates for all participants. However, this is unclear from the ms.

P9 L174: "These categories were chosen as they are represented in different regions of the brain [20], and ensuring there are no generalization effects between the CS's during learning." Ensuring seems bold. Perhaps: "...in different regions of the brain, in order to minimize generalization effects.." What generalization effects are you trying to minimize?

P11: I am confused by the "target" and "filler" structure. It is used to "ensure trials of each condition of interest were of equal distance to each other". Do you mean in order to make the same trial not appear many times in a row?

Target trials are non-reinforced trials. However, filler trials are not all reinforced. So I guess that some filler trials are omitted and classified as filler trials in order to make an equal amount of trials for the different conditions. Is this true? If so, please state in the ms. Supp mat 4 also confuses me as it depicts block of trials. Is a block of target trials 7 trials long? Does Sup mat 4 depict the actual order of blocks? In that case the experiment actually starts with a long habituation period during which no shocks are delivered. How many trials are in a filler block? Threat learning should have $13 * 4 = 52$ trials. How are these divided on the different blocks? I may be missing something obvious, but I am confused about this.

P13 L263: "...discarded and using the linear point..." What does this mean? Which point?

P14 L283: Please define "ToM network".

P15: Regression models.

I have trouble understanding this. I guess you are trying to predict the similarity from trial to trial within your ROI:s by a regression matrices representing the different trials ordered into your

regressions in order to represent the different effects that you are interested in. However, the description confuses me.

"...we created template regression matrices for each feature of interest (figure 3 b) with elements i and j , each representing a trial in the experiment [23], where the correlation between the same stimuli would be represented at $i = j$ indices of the matrix. Each matrix thus consisted of 24×24 elements (one for each target trial of each condition).

Please specify "feature of interest", or state that you will have different for each period of your trials.

Don't you mean that $i=j$ indices represent the correlation between the same stimuli at the same trial? Indices i and j represent trials. So $(i,j+1)$ will represent the correlation between consecutive trials for the same stimuli, the way they are ordered in fig 3b.

"Each matrix thus consisted of 24×24 elements (one for each target trial of each condition)." Please define "target trial".

P16 EARLY ANTICIPATION PERIOD.

Here are 4 regressors created, some which are not depicted in fig 3b. These are the "feature of interest" for the early anticipation period?

About the "intentionality increase in time" and "Unintentionality increase in time". Intentionality is learned at the start of the experiment. Thus, neither intentionality or unintentionality can increase over time. However, the participants feelings or responses to the faces of the confederate can of course change over the experiment, but these regressors seem badly named. Please explain.

L 335 "Note that we did not conduct the analysis using a "decrease" of correlational values for faces for the extinction phase, since we saw no pattern formation during learning." What pattern formation are you referring to here?

L 337 "Instead, we chose to only report the regressors for increase of correlation, which represent a novel representation of safety during extinction." How can we be sure of that?

P17: CHOSEN OPTION PERIOD

Again I am confused by the regressors. Why not include a CS+ correlation increase over trials, which in my mind might represent the development of a conditioned response? Instead the change over trials are surmised to interact with intentionality.

Also, as before, I don't see how intentionality or unintentionality can develop over time as it is learned from the outset.

P18: L360- About extinction. Is regressor 6 and 7 added to the ones before or do they replace regressor 4 and 5 on page 17?

Results:

L 394: "Participants reported higher shock expectancy ($F(1,24) = 8.74, p = 0.007, \eta^2 = 0.26$) and receiving more shocks from the intentional vs. unintentional choices, irrespective of the outcome (figure 2 a)"

Why is this an ANOVA? Are both CS+ and CS- lumped together? What not a t-test between CS+ (intent) and CS+ (no intent)?

L 398: "Overall, participants thought CS+'s were chosen more often, regardless of intentionality." Statistics?

L 409: "Five participants reported doubting more than 50% of the time they spent engaged in the social interaction (for the whole distribution see electronic supplementary material 5 a). When added to the above reported statistical analyses, we found that the credibility did not affect any of the results reported above except for..."

So five participants were removed from the behavioral results. This is not reported under participants in the methods section. Were they removed in any other analyses?

L 428-429 The pupillometry results were insignificant, but $p < 0.001$ in reference [1]. Do you have any ideas why? It is a big difference. From the psychophysiological data it appears you have achieved contingency awareness, but not fear conditioning.

L434: "This allowed us to formulate specific predictions regarding the timeline of the integration of our two features of interest: social information and learned threat value." Please specify these prediction. Do you mean the formulation of regressor matrices?

L480 Please define the abbreviation "FFA".

L 489-498 One of the reasons for this study is "...to replicate these findings in a novel dataset...". Here is where this is attempted. The authors conclude that "Despite being weaker, our results are consistent with previous findings of neural pattern formation in response to threat learning, where an increase in correlations were observed as a main effect of CS type in the ACC, the insula [1] and the vmPFC." However, in [1] the CS+ (intent) appeared to differ from the other conditions. In the present study, according to the one graph that is shown in figure (3a), it appears both CS+ differ from both CS-. Does this possibly change the authors conclusion of replication? Does it look the same in the ACC and vmPFC that did not survive FDR correction. Oh wait, this is what is mentioned in the discussion on L 540? "We could not, however, replicate the increase in neural pattern correlations specific to the intentional threatening actions." Perhaps this could be mentioned also here?

Around here, please refer to figure 3a. As it stands, it is not referred to in the text.

L500- "For the anticipation period (i.e., the viewing of a face, before a choice was made) in the extinction phase we chose to test only for an increase in correlations (e.g., reflecting safety learning), rather than a decrease (e.g., reflecting a decline in threat), because neural correlation patterns in this anticipation phase failed to show evidence of threat learning (i.e., an increase in pattern similarity) in the first place."

Where is this failure to show threat learning reported?

Discussion:

I have a hard time evaluating the conclusions with all my questions on the data analysis unanswered. It appears the RSM and regression results are not discussed?

===PREPARING YOUR MANUSCRIPT===

===PREPARING YOUR REVISION IN SCHOLARONE===

- Any electronic supplementary material (ESM).
- If you are requesting a discretionary waiver for the article processing charge, the waiver form must be included at this step.
- If you are providing image files for potential cover images, please upload these at this step, and inform the editorial office you have done so. You must hold the copyright to any image provided.
- A copy of your point-by-point response to referees and Editors. This will expedite the preparation of your proof.

- Ensure that your data access statement meets the requirements at <https://royalsociety.org/journals/authors/author-guidelines/#data>. You should ensure that you cite the dataset in your reference list. If you have deposited data etc in the Dryad repository, please include both the 'For publication' link and 'For review' link at this stage.
- If you are requesting an article processing charge waiver, you must select the relevant waiver option (if requesting a discretionary waiver, the form should have been uploaded at Step 3 'File upload' above).
- If you have uploaded ESM files, please ensure you follow the guidance at <https://royalsociety.org/journals/authors/author-guidelines/#supplementary-material> to include a suitable title and informative caption. An example of appropriate titling and captioning may be found at https://figshare.com/articles/Table_S2_from_Is_there_a_trade-off_between_peak_performance_and_performance_breadth_across_temperatures_for_aerobic_scope_in_teleost_fishes_/3843624.

Author's Response to Decision Letter for (RSOS-202116.R0)

See Appendix A.

RSOS-202116.R1 (Revision)

Review form: Reviewer 3

Is the manuscript scientifically sound in its present form?

Yes

Are the interpretations and conclusions justified by the results?

Yes

Is the language acceptable?

Yes

Do you have any ethical concerns with this paper?

No

Have you any concerns about statistical analyses in this paper?

Yes

Recommendation?

Major revision is needed (please make suggestions in comments)

Comments to the Author(s)

The authors studied the neural substrates of threat learning in the context of social interactions using representational similarity analysis. They report the involvement of the insula and inferior frontal gyrus in the learning of threat contingencies. The study is based on an elegant paradigm, the topic is interesting and the manuscript rather is clear. While the fMRI analyses are overall well executed, I have serious concerns about the theoretical modelling of the RSA regression templates. The results are also quite underwhelming (e.g. no intentionality effect found), but I believe this might be due to the misconceiving RSA templates and can be improved. Indeed, the overall RSA pipeline is well implemented, but I do not agree with the logic of the regression templates (see below for specific comments). I strongly suggest to reconsider this aspect of the manuscript or to explain in much details why the authors believe their implementation of the regression templates is correct. For this reason, I recommend major revisions.

Major concern:

The only serious concern I have is regarding the implementation of the RSA., In particular, I believe the definition of regression templates is misconceived and should be reformulated.

In general, I believe that the relation between concepts of pattern formation, pattern similarity or increase in pattern similarity and learning for threat and social interactions should be better explained throughout the manuscript. The authors write "We formulated the regression matrices to predict an increase in trial-by-trial pattern correlations to increase in response to certain stimuli but not others." or "The regressor CS+ > CS- predicts trial-by-trial similarity to be higher for threat choices regardless of intentionality, than safe ones.", but it is not clear from a neurocomputational or neurofunctional perspective what this means.

To my understanding (and I might be wrong), the authors would like to identify among the candidates brain region, which ones are able to:

1. detect/classify situations with intentional vs unintentional social interactions (intentional > unintentional)
2. detect/classify the situations where a threat is expected (CS+ > CS-)
3. detect/classify the situations where a threat is expected only in the intentional context AND with a gradual learning effect for the detection/classification (CS+intentional > CSother)
4. detect/classify the situations where a threat is expected only in the unintentional context AND with a gradual learning effect for the detection/classification (CS+unintentional > CSother)

The RSA framework can be compared to classification where each pattern is conceived as a "representation" with some information stored. A typical classifier, which is able to discriminate between two categories, is conceptualized in RSA by a similarity matrix where pairs of trials of the same category have high similarity (e.g. intentional-intentional pairs or unintentional-unintentional pairs) and pairs of trials of different category have low similarity (e.g. intentional-unintentional pairs). The proposed regression templates do not follow this logic.

For example, in the intentional > unintentional template, the authors state that such template predicts higher similarity for intentional pairs of trials than for unintentional ones. However, it is not clear how this relates to a brain regions detecting intentionality vs unintentionality in social interactions. Based on this template (intentional > unintentional), the proposed properties can be summarized as follow: 1) activity patterns during all intentional trials are very similar to each (similarity of 1), as such the same "information" is processed by this brain region for all

intentional trials; 2) activity patterns during unintentional trials are anti-correlated (similarity of -1) and each pattern is somewhat unique, as such different “information” is processed by this brain region for each unintentional trials; 3) activity patterns for intentional-unintentional pairs of trials are unrelated (similarity of 0). To my understanding, this template does not implement a brain region capable of detecting intentionality vs unintentionality in social interactions, and I believe the following template would be more appropriate: 1) intentional pairs of trials should exhibit similar patterns (similarity of 1 in the top left triangle); 2) unintentional pairs of trials should also exhibit similar patterns (similarity of 1 in the bottom right triangle); 3) intentional-unintentional pairs of trials should exhibit different patterns (similarity of 0 in the bottom left quadrant). The same logic should be applied to all templates. In addition to reformulating the templates, I also recommend to further detail the expected neural properties associated with each template in terms of activity patterns and ability to detect/classify different types of stimuli (which is the core of the RSA framework). This will help the reader to understand the relevance of each regression template.

I would like to note that I might be simply missing the relevance of the current models for predicting threat learning in social context, and how this translates into correlated or anti-correlated patterns. In this case, it is important that the authors present these concepts more clearly and explain in details the rationale for their templates.

I see also other few problematic points with the current templates:

- First, from a mathematical perspective, the case of anti-correlated patterns (similarity of -1) is problematic. For example in the intentional > unintentional template, all pairs of unintentional trials have a similarity of -1. Mathematically, it is impossible to conceive 12 different patterns that will all be anti-correlated to each other.
- Second, the repeated presentation of the same stimuli is in general expected to lead to rather similar activity patterns for relevant brain regions, so same-category pairs of trials should not have low similarity or anti-correlation (e.g. in the intentional > unintentional template, the same-category unintentional pairs of trials are also set to -1 although the exact same stimuli was presented).

Minor comments:

- methods, page 15, lines 328-330: “Each matrix consisted of 24×24 elements, each element representing the similarity between the target trials (7×4 , 7 trials and 4 conditions)”, $7 \times 4 = 28$.
- methods, page 21, lines 449-452: the group-level statistics should be performed on the individual weights rather than the associated individual t-values (as in univariate GLM analyses where beta estimates from the single subject level are fed into the group level GLM).
- methods, page 21, lines 455-457: If the trial-by-trial correlation matrices are Fischer-transformed, I believe the regression templates should also be Fischer-transformed for the regression.
- results, pages 28-30: it is not clear whether the reported ROIs with significant fitting were found in the right or left hemispheres, please specify.
- figure 3: the bar charts and legends have different sizes across the different panels.

Decision letter (RSOS-202116.R1)

Dear Ms Undeger

The Editors assigned to your paper RSOS-202116.R1 "Model-based representational similarity analysis of BOLD fMRI captures threat learning in social interactions" have now received comments from reviewers and would like you to revise the paper in accordance with the reviewer comments and any comments from the Editors. Please note this decision does not guarantee eventual acceptance.

Please submit your revised manuscript and required files (see below) no later than 21 days from today's (ie 02-Sep-2021) date. Note: the ScholarOne system will 'lock' if submission of the revision is attempted 21 or more days after the deadline. If you do not think you will be able to meet this deadline please contact the editorial office immediately.

on behalf of Prof Essi Viding (Subject Editor)
openscience@royalsociety.org

Reviewer comments to Author:

Reviewer: 3

Comments to the Author(s)

The authors studied the neural substrates of threat learning in the context of social interactions using representational similarity analysis. They report the involvement of the insula and inferior frontal gyrus in the learning of threat contingencies. The study is based on an elegant paradigm, the topic is interesting and the manuscript rather is clear. While the fMRI analyses are overall well executed, I have serious concerns about the theoretical modelling of the RSA regression templates. The results are also quite underwhelming (e.g. no intentionality effect found), but I believe this might be due to the misconceiving RSA templates and can be improved. Indeed, the overall RSA pipeline is well implemented, but I do not agree with the logic of the regression

templates (see below for specific comments). I strongly suggest to reconsider this aspect of the manuscript or to explain in much details why the authors believe their implementation of the regression templates is correct. For this reason, I recommend major revisions.

Major concern:

The only serious concern I have is regarding the implementation of the RSA. In particular, I believe the definition of regression templates is misconceived and should be reformulated.

In general, I believe that the relation between concepts of pattern formation, pattern similarity or increase in pattern similarity and learning for threat and social interactions should be better explained throughout the manuscript. The authors write "We formulated the regression matrices to predict an increase in trial-by-trial pattern correlations to increase in response to certain stimuli but not others." or "The regressor CS+ > CS- predicts trial-by-trial similarity to be higher for threat choices regardless of intentionality, than safe ones.", but it is not clear from a neurocomputational or neurofunctional perspective what this means.

To my understanding (and I might be wrong), the authors would like to identify among the candidates brain region, which ones are able to:

1. detect/classify situations with intentional vs unintentional social interactions (intentional > unintentional)
2. detect/classify the situations where a threat is expected (CS+ > CS-)
3. detect/classify the situations where a threat is expected only in the intentional context AND with a gradual learning effect for the detection/classification (CS+intentional > CSother)
4. detect/classify the situations where a threat is expected only in the unintentional context AND with a gradual learning effect for the detection/classification (CS+unintentional > CSother)

The RSA framework can be compared to classification where each pattern is conceived as a "representation" with some information stored. A typical classifier, which is able to discriminate between two categories, is conceptualized in RSA by a similarity matrix where pairs of trials of the same category have high similarity (e.g. intentional-intentional pairs or unintentional-unintentional pairs) and pairs of trials of different category have low similarity (e.g. intentional-unintentional pairs). The proposed regression templates do not follow this logic.

For example, in the intentional > unintentional template, the authors state that such template predicts higher similarity for intentional pairs of trials than for unintentional ones. However, it is not clear how this relates to a brain regions detecting intentionality vs unintentionality in social interactions. Based on this template (intentional > unintentional), the proposed properties can be summarized as follow: 1) activity patterns during all intentional trials are very similar to each (similarity of 1), as such the same "information" is processed by this brain region for all intentional trials; 2) activity patterns during unintentional trials are anti-correlated (similarity of -1) and each pattern is somewhat unique, as such different "information" is processed by this brain region for each unintentional trials; 3) activity patterns for intentional-unintentional pairs of trials are unrelated (similarity of 0). To my understanding, this template does not implement a brain region capable of detecting intentionality vs unintentionality in social interactions, and I believe the following template would be more appropriate: 1) intentional pairs of trials should exhibit similar patterns (similarity of 1 in the top left triangle); 2) unintentional pairs of trials should also exhibit similar patterns (similarity of 1 in the bottom right triangle); 3) intentional-unintentional pairs of trials should exhibit different patterns (similarity of 0 in the bottom left quadrant). The same logic should be applied to all templates. In addition to reformulating the templates, I also recommend to further detail the expected neural properties associated with each template in terms of activity patterns and ability to detect/classify different types of stimuli (which is the core of the RSA framework). This will help the reader to understand the relevance of each regression template.

I would like to note that I might be simply missing the relevance of the current models for predicting threat learning in social context, and how this translates into correlated or anti-correlated patterns. In this case, it is important that the authors present these concepts more clearly and explain in details the rationale for their templates.

I see also other few problematic points with the current templates:

- First, from a mathematical perspective, the case of anti-correlated patterns (similarity of -1) is problematic. For example in the intentional > unintentional template, all pairs of unintentional trials have a similarity of -1. Mathematically, it is impossible to conceive 12 different patterns that will all be anti-correlated to each other.
- Second, the repeated presentation of the same stimuli is in general expected to lead to rather similar activity patterns for relevant brain regions, so same-category pairs of trials should not have low similarity or anti-correlation (e.g. in the intentional > unintentional template, the same-category unintentional pairs of trials are also set to -1 although the exact same stimuli was presented).

Minor comments:

- methods, page 15, lines 328-330: "Each matrix consisted of 24×24 elements, each element representing the similarity between the target trials (7×4 , 7 trials and 4 conditions)", $7 \times 4 = 28$.
- methods, page 21, lines 449-452: the group-level statistics should be performed on the individual weights rather than the associated individual t-values (as in univariate GLM analyses where beta estimates from the single subject level are fed into the group level GLM).
- methods, page 21, lines 455-457: If the trial-by-trial correlation matrices are Fischer-transformed, I believe the regression templates should also be Fischer-transformed for the regression.
- results, pages 28-30: it is not clear whether the reported ROIs with significant fitting were found in the right or left hemispheres, please specify.
- figure 3: the bar charts and legends have different sizes across the different panels.

===PREPARING YOUR MANUSCRIPT===

===PREPARING YOUR REVISION IN SCHOLARONE===

Author's Response to Decision Letter for (RSOS-202116.R1)

See Appendix B.

RSOS-202116.R2

Review form: Reviewer 3

Is the manuscript scientifically sound in its present form?

Yes

Are the interpretations and conclusions justified by the results?

Yes

Is the language acceptable?

Yes

Do you have any ethical concerns with this paper?

No

Have you any concerns about statistical analyses in this paper?

No

Recommendation?

Accept with minor revision (please list in comments)

Comments to the Author(s)

This is the second time I review this manuscript. The authors have satisfactorily addressed my main concerns. I have a couple of remaining minor issues that should be addressed/discussed.

minor concerns:

- One of the main topic addressed by the authors is the context of intentional vs unintentional social interactions. They could not replicate previous results about intentionality and only wrote: "Although we replicated the behavioural effects of intentionally (1), we did not observe the expected neural effects, which might be related to a lack of power and/or methodological differences as compared to previous work." The authors should develop this

paragraph to provide better explanations about the lack of results for the intentionality effect, which is conceived as a major aspect of the experimental design.

- A possible way to address the lack of conclusive results about intentionality would be to explore other ROIs. In the present and previous work from the authors, they select a priori ROIs based on the literature. I would recommend exploring a bit more the data by applying a wholebrain approach to the RSA or by using the same data with a univariate GLM approach to identify the ROIs in which the BOLD signal varies significantly as a function of intentionality and to further apply RSA within these ROIs (no problem of double dipping with univariate and multivariate analyses applied to the same data). Although restricting the number of tested ROIs is in principle a good approach, I believe in this case, with a novel paradigm, a more explorative approach could be justified.

Decision letter (RSOS-202116.R2)

Dear Ms Undeger

On behalf of the Editors, we are pleased to inform you that your Manuscript RSOS-202116.R2 "Model-based representational similarity analysis of BOLD fMRI captures threat learning in social interactions" has been accepted for publication in Royal Society Open Science subject to minor revision in accordance with the referees' reports. Please find the referees' comments along with any feedback from the Editors below my signature.

Please submit your revised manuscript and required files (see below) no later than 7 days from today's (ie 11-Oct-2021) date. Note: the ScholarOne system will 'lock' if submission of the revision is attempted 7 or more days after the deadline. If you do not think you will be able to meet this deadline please contact the editorial office immediately.

on behalf of Essi Viding (Subject Editor)
 openscience@royalsociety.org

Associate Editor Comments to Author:

Comments to the Author:

Thank you for engaging with the reviewer's concerns - there are a number of remaining matters to be addressed, but the paper certainly appears to be on the right track. Please carefully respond to the remaining comments in your revision - do make sure you also include your original files etc. (the decision email includes the workflow requirements).

Reviewer comments to Author:

Reviewer: 3

Comments to the Author(s)

This is the second time I review this manuscript. The authors have satisfactorily addressed my main concerns. I have a couple of remaining minor issues that should be addressed/discussed.

minor concerns:

- One of the main topic addressed by the authors is the context of intentional vs unintentional social interactions. They could not replicate previous results about intentionality and only wrote: "Although we replicated the behavioural effects of intentionally (1), we did not observe the expected neural effects, which might be related to a lack of power and/or methodological differences as compared to previous work." The authors should develop this paragraph to provide better explanations about the lack of results for the intentionality effect, which is conceived as a major aspect of the experimental design.
- A possible way to address the lack of conclusive results about intentionality would be to explore other ROIs. In the present and previous work from the authors, they select a priori ROIs based on the literature. I would recommend exploring a bit more the data by applying a wholebrain approach to the RSA or by using the same data with a univariate GLM approach to identify the ROIs in which the BOLD signal varies significantly as a function of intentionality and to further apply RSA within these ROIs (no problem of double dipping with univariate and multivariate analyses applied to the same data). Although restricting the number of tested ROIs is in principle a good approach, I believe in this case, with a novel paradigm, a more explorative approach could be justified.

===PREPARING YOUR MANUSCRIPT===

===PREPARING YOUR REVISION IN SCHOLARONE===

<https://royalsociety.org/journals/authors/author-guidelines/#data>. You should ensure that

you cite the dataset in your reference list. If you have deposited data etc in the Dryad repository, please only include the 'For publication' link at this stage. You should remove the 'For review' link.

Author's Response to Decision Letter for (RSOS-202116.R2)

See Appendix C.

Decision letter (RSOS-202116.R3)

Dear Ms Undeger,

I am pleased to inform you that your manuscript entitled "Model-based representational similarity analysis of BOLD fMRI captures threat learning in social interactions" is now accepted for publication in Royal Society Open Science.

The proof of your paper will be available for review using the Royal Society online proofing system and you will receive details of how to access this in the near future from our production office (openscience_proofs@royalsociety.org). We aim to maintain rapid times to publication after acceptance of your manuscript and we would ask you to please contact both the production office and editorial office if you are likely to be away from e-mail contact to minimise delays to

publication. If you are going to be away, please nominate a co-author (if available) to manage the proofing process, and ensure they are copied into your email to the journal.

on behalf of Prof Essi Viding (Subject Editor)
openscience@royalsociety.org

Appendix A

Associate Editor Comments to Author (Dr Jonathan Roiser):

Associate Editor: 1

Comments to the Author:

Thank you for submitting your work to RSOS. Your manuscript has been reviewed by two experts in the field, who felt that the work was of interest but also raised several concerns and queries. We would be willing to consider a revised version that addresses these points, but at this stage cannot guarantee that the manuscript will be accepted, as some of the comments are quite critical. Please ensure that all reviewer queries are addressed in a point-by-point fashion in your revised submission.

Authors' response:

Thank you for the opportunity to revise and submit our manuscript. We want to express our gratitude for the constructive feedback we received from both you and the reviewers. We have now addressed all points raised by the reviewers, and hope that you agree that our revised manuscript is improved. Below, we list the reviewers' comments in italics, followed by our responses.

Reviewer comments to Author:

Reviewer: 1

Comments to the Author(s)

Summary:

In the current study, the authors explored how people perceive and learn about threatening actions of others. They explored the behavioral, physiological, and neural responses to threat learning by watching confederates make decisions which intentionally or unintentionally resulted in a shock to participants. Importantly, this study attempted to directly replicate an earlier study (published in Cerebral Cortex) from the same group. The only difference is that a different sample of participants tested, and additional analyses were done in order to reveal the underlying aspects of the social interactions in the experiment. Although the behavioral results supported their hypotheses and replicated their earlier findings, the pupillometry and neuroimaging results did not match up as well. It was interesting to see the pupillometry portion of the study,

and the representational similarity analyses was very thoroughly implemented. However, there are a number of critiques that need to be addressed.

I didn't realize that this was a near-exact replication until I looked at the author's previous study published in Cerebral Cortex and saw the task and stimuli. This is not a bad thing at all! However readers will want to know how similar this is to previously-published study. The authors should be more explicit about this, in the abstract, the methods, and discussion.

Also, the RSA technique is said to be novel but it probably is not.

Authors' response:

We would like to thank the reviewer for agreeing with the potential interest in our manuscript. We added a table in the electronic supplementary materials that directly compares the differences between the two studies in methodology and results. Electronic Supplementary Material 13 now reads:

Supplementary Material 13: Table comparing differences between Undeger et al., 2020 (1) and the current manuscript. RSA effects are reported as a result of an ANOVA for the 2020 study, and multiple regression for the current study. Only significant findings are discussed.

		Study	
		2020	Current
Study design	Timing of shocks	Before CS onset	200 ms after CS onset
	Phases of the experiment	Aversive Learning	Aversive Learning and Extinction Learning
Behavioral effects	Reported number of stimuli chosen by confederates	Intentional > Unintentional	CS+>CS-
	Decrease in likability	Intentional > Unintentional	-
	No. of participants that doubted the manipulation (>50%)	11	5
Pupil response	Sample size	35	23
	Significant effects	CS+ > CS-	-
RSA effects (per brain region)	Sample size	33	26
	ACC	CS+ > CS- and Intentional > Unintentional	-
	IFG	CS+ > CS- and Intentional > Unintentional	CS+>CS-
	Insula	CS+ > CS- and Intentional > Unintentional	CS+>CS-
	vmPFC	CS+>CS-	-
	arSTS	CS+ > CS- and Intentional > Unintentional	-
	dmPFC	CS+ > CS- and Intentional > Unintentional	-

Plus, we mentioned the differences between the two studies in the abstract, the methods and discussion.

In the abstract, lines 41-47 now reads: “Our previous work suggested that the intentionality of a threatening action leads to an increase in neural pattern correlations in brain regions related to intentionality. Surprisingly, we found no effects of intentionality at the neural level in this study, even though behavioural measures showed an inflation of threat outcomes (i.e. the number of shocks received) when they seemed intentional. We discuss if and how this experimental design can be utilized in the future.”

In the Methods section, line 257-259 now reads as: “Note that in our previous work (1), shocks were delivered right before the presentation of the CS+ images.”

Based on the reviewers suggestions, we have now extensively revised the Discussion (line 804 - 1028) and we hope that we now have fully addressed the reviewer’s concerns.

Critiques:

Methodological

1. I would like a citation for why these ToM ROIs that were selected. STS seems like an odd choice, given its strong role in basic language processing. Why not the ATL instead? It has been linked to person knowledge and ToM for over 20 years.

Authors’ response:

The reviewer is correct in that the ATL has been often reported in research on ToM. Yet, in the interest of limiting the number of multiple comparisons, we had to make a few difficult choices between ROI’s considered standard in the field of social cognition and theory of mind (ToM). We selected the superior temporal sulcus (STS), because it has repeatedly been reported in tasks related to social cognition, such as ToM (1), perspective taking (2), and specifically intention prediction (3) in humans and non-human primates (4).

Importantly, we did not want to limit our investigation to regions reported in studies using univariate fMRI analyses only, and were inspired by research using multivariate techniques (5,6). Additionally, STS is one of the regions that integrate mentalizing and autobiographical memory (7), which we believe is a good fit to the learning paradigm we implemented. Finally, we wanted to take into account the fact that our

study involves a deception of “online interaction” and STS has been reported in a live face-to-face interaction study (8).

Due to space constraints in the main text, we were unfortunately unable to provide detailed literature about each ROI. We would be happy to add additional information about the STS in the main text, should the reviewer think it would strengthen the manuscript.

2. Are the two vmPFC ROIs the same for the threat learning and ToM analyses? A visual depiction of the ROIs would be helpful.

Authors’ response:

The two vmPFC ROI’s are the same. In fact, analyses for threat learning and ToM were not conducted separately but were only modelled as separate regressors in the same regression model. A visual depiction of the ROI’s has been added to the main text as figure 2 (line 353):

Figure 2. Visualisation of regions of interest used as masks in the representational similarity analysis, presented on the standard MNI brain used for the analyses.

3. It is stated several times that this regression-based RSA is novel. How does it differ from previous uses (Tamir et al., 2016; Brooks & Freeman, 2018; Stolier & Freeman, 2016; Lee Masson et al., 2018; Bracci et al., 2015)? See Popal et al., 2019 for a review of RSA studies that used multiple regression. If, as I suspect, the method is not novel, please alter the text to reflect this.

Authors' response:

Our use of the term “novel” was unfortunate as we meant to say that the technique has been applied to single-trial RSA study in humans for the first time. Additionally, we modelled regressors that correct for the similarity of a stimulus item to itself, such as the “category” target regressor (figure 4 b) that we believe are not commonly used in the current literature. We thank the reviewer for providing literature using the same method. We have revised the text and removed mentions of the method being novel. Since these changes are spread throughout the text, we do not include them all in this letter. Instead, they are marked as deletions in the edited version of the manuscript.

4. It is good that the ITI is long, but if the trials were not randomized couldn't there still be an increase in false positives, especially if within-run comparisons were made (Mumford et al., 2014)? Were multiple runs acquired and were trial-by-trial comparisons done between runs to somewhat alleviate this issue?

Authors' response:

Indeed, temporal correlations in the data may strongly influence the similarity between response patterns, and if not taken into account, result in false positives. Full randomization and between-run comparisons were not possible because of the nature of the task, which aimed to assess aversive learning with relatively few trials. However, the experiment used a previously validated design to control for this issue (9,10):

Firstly, as the reviewer suggests, we used long ITI's to ensure there is enough distance between the trials. Second and crucially, with the “target” and “filler” trial structure (see electronic supplementary material 4), we make sure that each correlation we report for a given trial pair in a condition is separated by the same temporal

distance. Namely, trial-to-trial correlations for each condition that we compare are always separated by 3 other target trials, plus an average of 0-6 filler trials. In the supplementary material example scheme, you can see that the first trial of CS_{+intent} and the second trial of CS_{+intent} have 3 trials between them, and so does the first trial of CS_{-intent} and the second trial of CS_{-intent}. This is true for all conditions. We do not directly compare target trials within a sequence (e.g., first trial of CS_{+intent} and first trial of CS_{+unintent}) as they are not separated by equal numbers of trials (i.e., varies between 0-2 other target trials). Instead, we compare the off-diagonal correlation values (i.e., the consecutive target trial correlations within a condition) between conditions.

Thus, we agree with the reviewer that there is temporal noise in the data, but argue that the noise does not affect the correlations we report in the manuscript. Importantly, the order of the stimuli is counter-balanced between participants, as are the images used as stimuli, meaning differences in stimuli or stimulus-order effects cannot explain the effects we observe either. We additionally tried to control stimulus effects by adding the “category” target template (figure 4 b) that models the similarity of a stimulus to itself throughout the experiment.

See also comments 4 – 10 by reviewer #2.

5. I would be useful to provide visual depiction of the ROIs to understand exactly what areas you chose as your ROIs (most researchers aren't very good at visualizing MNI coordinates).

Authors' response:

A visual depiction of the ROI's has been added to the main text as figure 2 (line 353), as mentioned in response to comment #2.

Discussion/interpretation

6. The IFG finding needs to be unraveled. The left IFG has been linked to language production for over 100 years. Please add text to the discussion. Note that the authors state that the IFG finding replicates Visser's study but I checked and Visser does not mention the IFG in their study.

Authors' response:

As suggested by the reviewer, we have added more information about the IFG, and discussed potential interpretations of our results in greater detail. As the changes we made are quite extensive we omit copying them here. Please find them between line 869 – 919 in the main text.

In the 2011 Journal of Neuroscience article (11) Visser et al. found an increase in trial-by-trial similarity in response to CS+ images. The reviewer is correct, wrong references were used in this text. We have fixed it to the correct reference as seen in line 803.

7. ToM is usually right lateralized (Corbetta et al., 2008; Boccadoro et al., 2019; Scholz et al., 2009). Laterality wasn't discussed.

Authors' response:

Since we did not find any effects in neural activity related to intentionality (i.e. ToM) in our study, we believe that discussing specific aspects of neural correlates of ToM (e.g. lateralization) would be beyond the scope of the current paper. Should the reviewer or editor, however, prefer us to discuss these aspects, we would be happy to add a section to the introduction.

8. The discussion is weak. It should discuss the current findings in greater detail. The current version is very much oriented towards discussing how the findings are similar or different from this group's prior study. I like this and it would actually help readers if this were to be put into table format. As a reader, I would like to know: should I use this task and if I do, what can I expect to find? However, you should also elaborate on the current findings for readers who are unaware of your Cerebral Cortex study.

Authors' response:

We would like to thank the reviewer for this comment. We have tried to incorporate the points mentioned in this comment in our updated discussion section line 804 – 1017. We provide an overview of limitations to consider if this task is to be used in future studies (line 925 – 1017), which now reads as: “Our replication efforts point to several important considerations that should be noted when implementing paradigms similar to ours in future studies. Firstly, future work should examine potential effects of the timing of aversive stimulation by manipulating the onset of the shock in relation

to the CS+. Secondly, stopping rules based on learning should be considered when recruiting participants. As we failed to replicate learning effects in pupillometry data, this should be considered as a potential factor in driving the difference between the results of the two studies. Finally, considering the well-documented effects of intentionality on pain and the involvement of the insula (13,32), future studies should consider using painful rather than uncomfortable aversive stimuli.”

We also provided a table of comparison which reports the differences between the two studies in design and outcomes (electronic supplementary material 13). We hope that the changes made addressed the reviewers concerns.

Minor:

1 - • I was confused why the authors called extinction learning “safety learning”. I have never heard of that term used before, whereas extinction learning is widely used. It was unclear why this distinction was made.

Authors’ response:

As our manuscript can be useful to different audiences, we tried to use as common terms as possible. Thus, we chose to call the phase of the experiment “extinction phase” but refer to the outcomes as “safety learning”. We chose this wording to highlight the fact that the participant is learning that the confederates’ choices (and thus the confederates themselves) are safe. As suggested by the reviewer, we have replaced the term “safety” with “extinction” to avoid confusion. As these changes are quite spread out, we refrain from repeating them here. They can be seen in highlighted as changes in the edited version of the manuscript.

2 - • When describe the scan acquisition parameters, the authors had a typo and I assume mixed TR with total scan time for the T1.

Authors’ response:

We thank the reviewer for pointing out this mistake. Line 326 now reads as: “(repetition time = 6.4 ms, ...”

3 - • The “BOLD-fMRI results” section seems to just be methods in how the RSMs are created and how RSA will be done. Perhaps it would make the paper and the RSA

easier to understand if you moved this to the methods section, with a subsection for RSA and RSM creation. Maybe a sub-subsection for each RSM?

Authors' response:

We thank the reviewer for helping us to make the description of our method clearer. Accordingly, we have made the suggested changes and moved the mentioned paragraphs to the Methods section. As the changes are quite extensive, we refrain from rephrasing them here. They can be found highlighted in line 375 - 569 of the manuscript.

4 - • *Figure 2 is hard to read. I recommend (1) decreasing the number of increments on the y-axis for (a) and (d)...it's crowded and hard to read ; (2) increase the resolution (it isn't very crisp); and (3) if allowed, use color to make it prettier.*

Authors' response:

We thank the reviewer for their suggestions. The revised version of figure 3 (previously figure 2) is:

5 - • *Figure 3 is too low resolution. It will look better in print if efforts are made to improve this.*

It would be better to provide the supplementary material document in a more accessible format, such as .doc so that people without Macs can open it. I was unable to open it on my PC.

Authors' response:

We did not submit the high-resolution image files for this first round of reviews. We apologize for any inconvenience this might have caused the reviewer. We provide high resolution images and a pdf version of the supplementary materials in this round.

Reviewer: 2**Comments to the Author(s)**

This paper attempts to replicate and extend a previous study using a threat learning paradigm where there are two confederates, one that can provide shocks (UCS) with intent, and another one, whose UCS is not provided with intent. Data is explored with RSM and a novel RSA regression method. Perhaps it is somewhat because of the tricky analyses, but I find myself with many questions and concerns when reading this paper. Foremost about the analyses and how to interpret the results. I hope by addressing these concerns, the authors will be helped in improving the readability of the paper.

Authors' response:

We thank the reviewer for providing us with comments that make our manuscript stronger. We hope that we were able to address all the issues and the reviewer finds the readability of the edited manuscript improved.

Methods:

1- P9 L162: "As an independent index of threat learning we collected pupillometry data, excluding participants who had more than 33% of trials of any condition missing (missing trial defined as > 50% missing sample for that trial) (n =3)"

At this point in the paper, the "conditions" are not defined. Which are they? CS+ (intent), CS- (intent), CS+ (no intent), CS- (no intent)?

Authors' response:

We thank the reviewer for pointing this out. Line 192 now reads: "... experimental condition (i.e. CS⁺_{intent}, CS⁺_{unintent}, CS⁻_{intent} and CS⁻_{unintent}) missing."

2 - I guess the same two individuals acted as confederates for all participants. However, this is unclear from the ms.

Authors' response:

We now added this information in the text. Line 210 now reads: "The same individuals acted as confederates for all participants."

3 - P9 L174: "These categories were chosen as they are represented in different regions of the brain [20], and ensuring there are no generalization effects between the CS's during learning." Ensuring seems bold. Perhaps: "...in different regions of the brain, in order to minimize generalization effects.." What generalization effects are you trying to minimize?

Authors' response:

We agree that the wording in the mentioned sentence is strong. We changed the wording and provided a reference in the main text to the generalization effects we were considering (12). The sentence in line 204 now reads as: "These categories were chosen as they are represented in different regions of the brain (13) in order to minimize generalization effects between the CS's during learning (21)." with the underlined changes. By using stimuli from different categories, we were trying to minimize the possible generalization effects that could reduce the differences in neural or pupil responses to the two CS+ stimuli during the extinction phase.

4 - P11: I am confused by the "target" and "filler" structure. It is used to "ensure trials of each condition of interest were of equal distance to each other". Do you mean in order to make the same trial not appear many times in a row?

Authors' response:

We realise that the choice for target and filler trials was not explained well. Our aim with target and filler structure was not to make the same trial not appear many times in a row, but instead 1) to control for temporal proximity, 2) to ensure that shock-related

activity could not confound CS-related activity, and 3) ensure that the trial order appeared random to the participant. Please see our response to comment 4 from reviewer #1 and to comment 5-7 from reviewer #2 below. What we tried to communicate with “*ensure trials of each condition of interest were of equal distance to each other*” was that the time between consecutive presentations of a stimulus type was the same across stimulus types. This is essential in paradigms using single-trial correlations, as the temporal distance between two stimuli can influence the strength of the pattern correlations (our measure of similarity). We have aimed to clarify this in the electronic supplementary material:

Electronic supplementary material 4 (page 5) now reads as: “With the “target” and “filler” trial structure (below), we make sure that each correlation we report for a given trial pair in a condition, is separated by the same temporal distance. Namely, the trial-to-trial correlations for each condition we compare are always separated by 3 other target trials, plus an average of 0-6 filler trials. In the trial scheme provided below, it can be seen that the first trial of CS_{+intent} and the second trial of CS_{+intent} have 3 trials between them, and so does the first trial of CS_{-intent} and the second trial of CS_{-intent}. Another advantage of the target and filler structure is that it prevents shock-related confounds in the analyses of the CS-related activation: the reinforced CS+ trials were all filler trials and thus discarded from the analysis. Note that we also had CS- filler trials in order to control for temporal proximity (as explained above), to make sure that all conditions had an equal number of trials, and also to make sure that the order of trials appeared random to the participant.”, novel additions underlined.

5 - Target trials are non-reinforced trials. However, filler trials are not all reinforced. So I guess that some filler trials are omitted and classified as filler trials in order to make an equal amount of trials for the different conditions. Is this true? If so, please state in the ms. Supp mat 4 also confuses me as it depicts block of trials.

Authors' response:

This is correct. While the CS- trials are indeed never reinforced, we applied the same distinction of filler and target trials a) to make sure that all conditions had an equal number of trials, but also b) to make sure that the order of trials appeared random to the participant. The sequence of target trials presents the stimuli in a fixed order, so

some shuffling is needed in between the sequences to prevent the sequence from becoming predictable. We hope that the information we added to the supplementary material in response to the comment #4 above made it clearer.

6 - Is a block of target trials 7 trials long?

Authors' response:

A block of “target” trials in this case is 4 trials long, including one trial of each type of experimental condition. As seen in the example below, “target” trials were blocks of trials that were identical throughout the experiment for a given participant. “Filler” trials, however, followed a pseudo-randomized order.

Example scheme: An example trial scheme for an experiment. “Target trials” highlighted in yellow, each number signifying a condition (e.g., 6 = CS⁺_{intent}, 0 = CS⁻_{intent}, 2 = CS⁺_{unintent}, 4 = CS⁻_{unintent}). Non-highlighted trials are the “filler trials”, each number signifying a condition. Please note that these lists were counterbalanced throughout the sample.

6, 0, 2, 4, 6, 0, 2, 4, 1, 5, 3, 3, 7, 1, 6, 0, 2, 4, 5, 7, 6, 0, 2, 4, 5, 7, 1, 3, 7, 5, 6, 0, 2, 4, 1, 1, 3, 7, 3, 5, 6, 0, 2, 4, 1, 7, 5, 3, 6, 0, 2, 4

7 - Does Sup mat 4 depict the actual order of blocks?

Authors' response:

Sup. Mat. 4 is an example of one of the trial orders a participant has used. As mentioned above in response to comment #6, this list is counter balanced throughout the participants.

8 - In that case the experiment actually starts with a long habituation period during which no shocks are delivered.

Authors' response:

The reviewer is correct, the experiment starts with two presentations of each trial, before the participant received information about the association between electrical

shocks and the CS+ stimuli. This is why in the model regressors learning is modelled as an increase that occurs only after the third trial.

9 - How many trials are in a filler block?

Authors' response:

We thank the reviewer for pointing out this mistake. The number reported in the manuscript is incorrect. The number of trials in a filler block varied between the blocks (see example scheme above). The total number of these trials were set to 6, leading to a 46% reinforcement rate throughout the experiment for all participants. Line 256 now reads as: "...delivery of shocks to the participant 46 percent of the time (6 out of 13) during threat..."

*10 - Threat learning should have $13 * 4 = 52$ trials. How are these divided on the different blocks? I may be missing something obvious, but I am confused about this.*

Authors' response:

The reviewer is correct that all trials add up to $13 * 4 = 52$. As mentioned above, each target trial has one of each type of stimulus condition and there are 7 blocks. Thus the experiment consists of 7 target and 6 filler trials for each condition. We hope that this is clearer after our mistake was corrected in response to comment #9 above.

11 - P13 L263: "...discarded and using the linear point..." What does this mean? Which point?

Authors' response:

While processing the pupillometry data, we used linear interpolation for the missing data points. We hope that the changes we made make it more clear in the text. Line 306 - 314 now reads as: "Part of the data cleaning for pupil data consisted of replacing missing values (e.g., because of blinking, scanner artefacts). Data around blinks (100 ms before and after each segment of missing values) were deemed unreliable and together with the missing segments replaced by linear trend between the preceding and following datapoints. However, trials that ended up with substantial signal loss

(more than 50% missing values after removing blink-related artefacts) were discarded entirely, and replaced using the linear trend between the preceding and following trial within that condition (to a maximum of 33% of trials per condition)".

12 - P14 L283: Please define "ToM network".

Authors' response:

We thank the reviewer for pointing this out. Line 127 now reads as: "...were selected based on their role in intentionality processing and theory of mind (ToM)." We also now refer to these regions as involved in "intentionality processing" as it is more directly related to our study. Line 336 now reads as: "For the regions related to intentionality processing, we...".

P15: Regression models.

13 - I have trouble understanding this. I guess you are trying to predict the similarity from trial to trial within your ROI:s by a regression matrices representing the different trials ordered into your regressions in order to represent the different effects that you are interested in. However, the description confuses me.

Authors' response:

The reviewer is correct, we created template matrices that modelled our hypotheses to predict trial-by-trial similarity in different regions. We hope that the change we made in the methods section of the manuscript can address the issues that led to confusion.

As these changes are quite large, we cannot mention them all here. Please refer to the section between line 375 - 568.

14 - "...we created template regression matrices for each feature of interest (figure 3 b) with elements i and j , each representing a trial in the experiment (23), where the correlation between the same stimuli would be represented at $i = j$ indices of the matrix. Each matrix thus consisted of 24×24 elements (one for each target trial of each condition).

Please specify "feature of interest", or state that you will have different for each period

of your trials.

Authors' response:

We agree that the “feature of interest” is vague in this sentence. By feature of interest we meant intentionality, aversive outcome, or the integration of both. We have carefully revised this section along with Minor comment #3 from Reviewer 1. We added the following sentence to the manuscript, line 398 – 400 : “We analysed the early anticipation and chosen option periods (figure 1 b) of each trial separately to capture responses to the interaction partners' faces and action outcomes, respectively. ”.

15 - Don't you mean that $i=j$ indices represent the correlation between the same stimuli at the same trial? Indices i and j represent trials. So $(i,j+1)$ will represent the correlation between consecutive trials for the same stimuli, the way they are ordered in fig 3b.

Authors' response:

Yes, the reviewer is correct. As mentioned above we made many changes to this part of the text based on the reviewers' comments. The corresponding section between line 379 – 393 now reads as: “Each matrix consisted of 24×24 elements, each element representing the similarity between the target trials (7×4 , 7 trials and 4 conditions). On the RSM's and regression templates, trials are grouped by condition for ease of visualization and modelling. Here, the diagonal line represents trial-by-trial similarity of each trial to itself (i,j) and is thus 1 for all RSM's (e.g. figure 4 a). The off-diagonal line immediately adjacent to the diagonal line ($i,j+1$) represents the similarity between consecutive trials and was the trial-by-trial similarity measure we used in previous research (1). Here, with the aid of the template regressors, we were able to model trials that are farther apart (e.g. $i, j+3$). Please refer to individual sections below for more information on how each template regressor was created.”

16 - “Each matrix thus consisted of 24×24 elements (one for each target trial of each condition).” Please define “target trial”.

Authors' response:

We added more information about target and filler trials in the text as suggested. Line 370 - 374 now reads: “As explained in the “threat learning” section above, RSM's were

created only of target trials, which were not reinforced, and were presented in identical order in blocks. These target blocks were presented together with filler blocks that did not have a fixed order and included the reinforced trials. Please refer to electronic supplementary material 4 for more details.”

P16 EARLY ANTICIPATION PERIOD.

17 - Here are 4 regressors created, some which are not depicted in fig 3b. These are the “feature of interest” for the early anticipation period?

Authors’ response:

Yes, the reviewer is correct. We hope that the changes we made based on the previous comment #14 about the “features of interest” also help make this part clear.

18 - About the “intentionality increase in time” and “Unintentionality increase in time”. Intentionality is learned at the start of the experiment. Thus, neither intentionality of unintentionality can increase over time. However, the participants feelings or responses to the faces of the confederate can of course change over the experiment, but these regressors seem badly named. Please explain.

Authors’ response:

The reviewer is correct, intentionality could not change in time. In accordance with the reviewer’s suggestion, we have now changed the names of these regressors in the main text. We refer to them as “Intentionality and CS+ integration”. As an example, line 493 now reads: “Intentionality and CS+ integration in time, ...”.

19 - L 335 “Note that we did not conduct the analysis using a “decrease” of correlational values for faces for the extinction phase, since we saw no pattern formation during learning.” What pattern formation are you referring to here?

Authors’ response:

We agree that this term has not been clear enough in the text. Line 507 – 511 now reads as: “Note that we did not conduct the analysis modelling a “decrease” of correlational values for faces for the extinction learning phase, since we didn’t capture any increase during learning.”.

20 - L 337 “Instead, we chose to only report the regressors for increase of correlation, which represent a novel representation of safety during extinction.” How can we be sure of that?

Authors’ response:

We did not see any significant effects on the template regressors that model an increase in pattern correlations for this period, thus we did not test the decrease in pattern correlations during the extinction phase since there was no increase during learning to begin with. Thus, we wanted to test if there were any novel patterns which might reflect a representation of safety during extinction. Line 510 - 511 now reads: “Instead, we chose to only report the regressors for increase of correlation, which might represent a novel representation of safety during extinction learning”.

P17: CHOSEN OPTION PERIOD

21 - Again I am confused by the regressors. Why not include a CS+ correlation increase over trials, which in my mind might represent the development of a conditioned response? Instead the change over trials are surmised to interact with intentionality. Also, as before, I don’t see how intentionality or unintentionality can develop over time as it is learned from the outset.

Authors’ response:

As the intentionality information is always present, as the reviewer points out, the only information that is collected over time by the participant is the outcomes of the confederate’s actions. Here, when an increase in neural correlations in response to intentional CS+ choices are modelled in contrast to unintentional ones, we can assume that the increase is representative of specifically the integration of intentionality with CS+ actions.

22 - P18: L360- About extinction. Is regressor 6 and 7 added to the ones before or do they replace regressor 4 and 5 on page 17?

Authors' response:

They replace the regressors 4 and 5. Line 534 now reads: "The following matrices were added to the regression model only for the threat learning phase:" and line 553 now reads: "The following template regression matrices were added to the regression model only for the extinction learning phase:".

We hope that the changes we made in the methods section made the explanation of how we used the regressors clearer. Please refer to line 375 - 568 in the main text.

Results:

23 - L 394: "Participants reported higher shock expectancy ($F(1,24) = 8.74, p = 0.007, \eta^2 = 0.26$) and receiving more shocks from the intentional vs. unintentional choices, irrespective of the outcome (figure 2 a)"

Why is this an ANOVA? Are both CS+ and CS- lumped together? What not a t-test between CS+ (intent) and CS+ (no intent)?

Authors' response:

We chose not to use a t-test in order to include the CS- responses, which accounts for the aversiveness of the stimuli. The ANOVA accounts for the 4-way interaction, and the main effects of Intentionality, and CS type. They were not lumped together. We agree that this was not communicated clearly. We add the information about the ANOVA. Line 598 - 600 now reads: "Participants reported higher shock expectancy (CS [2] x Intentionality [2], $F(1,24) = 8.74, p = 0.007, \eta^2 = 0.26$) and receiving more shocks from the intentional vs. unintentional choices, irrespective of the outcome (figure 3 a) (CS [2] x Intentionality [2], $F(1,24) = 4.72, p = 0.047, \eta^2 = 0.16$) ..."

24 - L 398: "Overall, participants thought CS+'s were chosen more often, regardless of intentionality."

Statistics?

Authors' response:

We have removed this sentence from the manuscript since this was only used as a summary sentence and might come off as a novel result as the reviewer suggest.

25 - L 409: "Five participants reported doubting more than 50% of the time they spent engaged in the social interaction (for the whole distribution see electronic supplementary material 5 a). When added to the above reported statistical analyses, we found that the credibility did not affect any of the results reported above except for..."

So five participants were removed from the behavioral results. This is not reported under participants in the methods section. Were they removed in any other analyses?

Authors' response:

All participants were included in all the analyses. This additional analysis was added for readers who might be interested in the effects of credibility of the online interaction on the behavioural results. Line 618 now reads: "Five participants out of the sample used for the analyses in this manuscript reported doubting more...".

26 - L 428-429 The pupillometry results were insignificant, but $p < 0.001$ in reference [1]. Do you have any ideas why? It is a big difference. From the psychophysiological data it appears you have achieved contingency awareness, but not fear conditioning.

Authors' response:

We were indeed puzzled by these results. We had successful learning outcomes as measured by pupillometry in pilot samples, as well as the reference (1). We used the same thresholds for data quality in both studies, thus the data quality in the reported sample should be comparable. However, we did have less power. In this study, we report pupillometry results for 23 out of 31 recruited participants where in the previous study we report 35 out of 40. Another difference between the two experiments is the timing of the electrical shocks. However, to our knowledge there is no previous literature to suggest that the timing in the current manuscript should be problematic, it is in fact the most commonly used "delay conditioning" design. We report differences in the

design and the outcomes of the current work and the reference (1) in electronic supplementary material 13.

27 - L434: *“This allowed us to formulate specific predictions regarding the timeline of the integration of our two features of interest: social information and learned threat value.”*

Please specify these prediction. Do you mean the formulation of regressor matrices?

Authors’ response:

We hope that the changes we made make this section clearer. Line 654 – 662 now reads as: “We formulated the regression matrices to predict an increase in trial-by-trial pattern correlations to increase in response to certain stimuli but not others. The regressor intentional > unintentional (figure 4 b) predicts trial-by-trial similarity to be higher for intentional choices regardless of aversiveness, than unintentional ones. The regressor CS+ > CS- predicts trial-by-trial similarity to be higher for aversive choices regardless of intentionality, than safe ones. The regressor CS+_{intentional} > CS_{other} predicts trial-by-trial similarity to be higher for intentional aversive choices, and to increase over time. The regressor CS+_{unintentional} > CS_{other} predicts trial-by-trial similarity to be higher for unintentional aversive choices, and to increase over time.”.

28 - L480 *Please define the abbreviation “FFA”.*

Authors’ response:

We thank the reviewer for bringing this to our attention. Line 337 now reads as: “We included fusiform face area (FFA) only when analysing the early anticipation period which consists of viewing the confederates’ faces.” .

29 - L 489-498 *One of the reasons for this study is “...to replicate these findings in a novel dataset...”. Here is where this is attempted. The authors conclude that “Despite being weaker, our results are consistent with previous findings of neural pattern formation in response to threat learning, where an increase in correlations were observed as a main effect of CS type in the ACC, the insula [1] and the vmPFC.” However, in [1] the CS+ (intent) appeared to differ from the other conditions. In the present study, according to the one graph that is shown in figure (3a), it appears both CS+ differ from*

both CS-. Does this possibly change the authors conclusion of replication? Does it look the same in the ACC and vmPFC that did not survive FDR correction. Oh wait, this is what is mentioned in the discussion on L 540? “We could not, however, replicate the increase in neural pattern correlations specific to the intentional threatening actions.” Perhaps this could be mentioned also here?

Authors’ response:

We now added that the specific increase in intentional aversive actions did not replicate. Additionally, as Reviewer 1 has suggested we provided a table with comparisons between the two studies in the electronic supplementary material 13. We hope that these two changes make it easier to compare the two studies and have an overview of the outcomes. Line 745 - 751 now reads as: “Despite being weaker, our results are consistent with previous findings of neural pattern formation in response to threat learning, where an increase in correlations were observed as a main effect of CS type in the ACC, the insula (1) (for a direct comparison between (1) and the current, see electronic supplementary material 13) and the vmPFC (9). We failed to replicate our previous findings which showed an increase of neural pattern correlations specific to intentional aversive choices in the IFG, the insula, the dmPFC, and the arSTS.”, with novel sections underlined.

30 - Around here, please refer to figure 3a. As it stands, it is not referred to in the text.

Authors’ response:

We thank the reviewer for pointing this out. Line 738 - 744 now reads as: “We observed greater trial-by-trial correlation in the ACC ($F(1,25) = 5.55, p = 0.03, \eta^2 = 0.18$), the insula ($F(1,25) = 17.04, p < 0.001, \eta^2 = 0.40$) (figure 4 a), the arSTS ($F(1,25) = 4.04, p = 0.05, \eta^2 = 0.14$), and the vmPFC ($F(1,25) = 5.72, p = 0.02, \eta^2 = 0.19$).”, with the underlined change.

31 - L500- “For the anticipation period (i.e., the viewing of a face, before a choice was made) in the extinction phase we chose to test only for an increase in correlations (e.g., reflecting safety learning), rather than a decrease (e.g., reflecting a decline in threat), because neural correlation patterns in this anticipation phase failed to show

evidence of threat learning (i.e., an increase in pattern similarity) in the first place.”

Where is this failure to show threat learning reported?

Authors' response:

We thank the reviewer for pointing this out. Line 726 - 727 now reads as: “No significant variance was found in any of the other regressor templates, for any other ROI.”

Discussion:

32 - I have a hard time evaluating the conclusions with all my questions on the data analysis unanswered. It appears the RSM and regression results are not discussed?

Authors' response:

We thank the reviewer for the valuable comments on the rest of the manuscript. We hope revised the discussion section quite extensively and hope that it discusses the RSM and regression results more thoroughly. The added discussion is quite long, thus we don't report it here. Please see main text line 803 - 1015.

References:

1. Koster-Hale J, Saxe R. Theory of Mind: A Neural Prediction Problem. *Neuron* [Internet]. 2013 Sep;79(5):836–48. Available from: <http://dx.doi.org/10.1016/j.neuron.2013.08.020>
2. Frith CD, Frith U. The Neural Basis of Mentalizing. Vol. 50, *Neuron*. 2006. p. 531–4.
3. Frith CD, Frith U. Mechanisms of Social Cognition. *Annu Rev Psychol* [Internet]. 2012 Jan 10;63(1):287–313. Available from: <http://www.annualreviews.org/doi/10.1146/annurev-psych-120710-100449>
4. Wittmann MK, Lockwood PL, Rushworth MFS. Neural Mechanisms of Social Cognition in Primates. *Annu Rev Neurosci*. 2018;41(1):99–118.
5. Koster-hale J, Richardson H, Velez N, Asaba M, Young L, Saxe R. Mentalizing regions represent distributed , continuous , and abstract dimensions of others ' beliefs. *Neuroimage* [Internet]. 2017;161(August):9–18. Available from: <http://dx.doi.org/10.1016/j.neuroimage.2017.08.026>

6. Young L, Cushman F, Hauser M, Saxe R. The neural basis of the interaction between theory of mind and moral judgment. *Proc Natl Acad Sci U S A*. 2007;104(20):8235–40.
7. Spreng RN, Mar RA. I remember you: A role for memory in social cognition and the functional neuroanatomy of their interaction. *Brain Res* [Internet]. 2012;1428:43–50. Available from: <http://dx.doi.org/10.1016/j.brainres.2010.12.024>
8. Redcay E, Dodell-Feder D, Pearrow MJ, Mavros PL, Kleiner M, Gabrieli JDE, et al. Live face-to-face interaction during fMRI: A new tool for social cognitive neuroscience. *Neuroimage* [Internet]. 2010;50(4):1639–47. Available from: <http://dx.doi.org/10.1016/j.neuroimage.2010.01.052>
9. Visser RM, Scholte HS, Beemsterboer T, Kindt M. Neural pattern similarity predicts long-term fear memory. *Nat Neurosci*. 2013;16(4):388–90.
10. Visser RM, Haan MIC De, Scholte HS, Kindt M. Trial-by-trial analysis of BOLD-MRI patterns uncovers the formation of associative fear memory : a protocol Trial - - - by - - - trial analysis of BOLD - - - MRI patterns uncovers the formation of associative fear memory : a protocol. 2016;(August).
11. Visser RM, Scholte HS, Kindt M. Associative Learning Increases Trial-by-Trial Similarity of BOLD-MRI Patterns. *J Neurosci* [Internet]. 2011 Aug 17;31(33):12021–8. Available from: <http://www.jneurosci.org/cgi/doi/10.1523/JNEUROSCI.2178-11.2011>
12. Dunsmoor JE, Murphy GL. Categories, concepts, and conditioning: how humans generalize fear. *Trends Cogn Sci* [Internet]. 2015;1–5. Available from: <http://dx.doi.org/10.1016/j.tics.2014.12.003>
13. Haxby J V., Hoffman EA, Gobbini MI. The distributed human neural system for face perception. Vol. 4, *Trends in Cognitive Sciences*. 2000. p. 223–33.
14. Undeger I, Visser RM, Olsson A. Neural Pattern Similarity Unveils the Integration of Social Information and Aversive Learning. *Cereb Cortex* [Internet]. 2020 Jun 4;1–10. Available from: <https://academic.oup.com/cercor/advance-article/doi/10.1093/cercor/bhaa122/5850962>

Appendix B

Reviewer: 3

Comments to the Author(s)

The authors studied the neural substrates of threat learning in the context of social interactions using representational similarity analysis. They report the involvement of the insula and inferior frontal gyrus in the learning of threat contingencies. The study is based on an elegant paradigm, the topic is interesting and the manuscript rather is clear. While the fMRI analyses are overall well executed, I have serious concerns about the theoretical modelling of the RSA regression templates. The results are also quite underwhelming (e.g. no intentionality effect found), but I believe this might be due to the misconceiving RSA templates and can be improved. Indeed, the overall RSA pipeline is well implemented, but I do not agree with the logic of the regression templates (see below for specific comments). I strongly suggest to reconsider this aspect of the manuscript or to explain in much details why the authors believe their implementation of the regression templates is correct. For this reason, I recommend major revisions.

Authors' Response:

Thank you for the constructive feedback and the opportunity to revise and re-submit our manuscript. We are thrilled to hear that you thought that we used an “elegant paradigm”, examined an “interesting topic”, and that “the manuscript rather is clear”. We hope that we were able to address all the remaining issues that the reviewer mentioned, and that the reviewer finds our manuscript improved.

Major concern:

- 1- The only serious concern I have is regarding the implementation of the RSA., In particular, I believe the definition of regression templates is misconceived and should be reformulated.*

In general, I believe that the relation between concepts of pattern formation, pattern similarity or increase in pattern similarity and learning for threat and social interactions should be better explained throughout the manuscript. The

authors write “We formulated the regression matrices to predict an increase in trial-by-trial pattern correlations to increase in response to certain stimuli but not others.” or “The regressor CS+ > CS- predicts trial-by-trial similarity to be higher for threat choices regardless of intentionality, than safe ones.”, but it is not clear from a neurocomputational or neurofunctional perspective what this means.

To my understanding (and I might be wrong), the authors would like to identify among the candidates brain region, which ones are able to:

- 1. detect/classify situations with intentional vs unintentional social interactions (intentional > unintentional)*
- 2. detect/classify the situations where a threat is expected (CS+ > CS-)*
- 3. detect/classify the situations where a threat is expected only in the intentional context AND with a gradual learning effect for the detection/classification (CS+intentional > CSother)*
- 4. detect/classify the situations where a threat is expected only in the unintentional context AND with a gradual learning effect for the detection/classification (CS+unintentional > CSother)*

Authors' response:

The reviewer is correct, we have created the intentional>unintentional template to detect the intentionality of choices; the CS+>CS- template to detect threat expectancy; the CS+_{Intentional} > CS_{other} template to detect a gradual learning effect for the intentional CS+, and the CS+_{unintentional} > CS_{other} template to detect a gradual learning effect for the unintentional CS+. We thank the reviewer for such an excellent summary. We added information about the regressors throughout the manuscript and hope that the reviewer finds the updated text clearer. In the introduction, line 127-131 now reads as: “On the neural level, we aimed to identify brain regions that are able to: i) detect intentional vs. unintentional social interactions (Intentional > Unintentional), ii) detect threat expectation (CS+ > CS-), iii) detect threat expectation only in the intentional context and with a gradual learning effect, as opposed to an intentional one (CS+_{intentional}> CS_{other} VS CS+_{unintentional} > CS_{other}).” In the Methods section line 374 now reads as: “Here, we aimed to detect learning about the intentional social partner.” Line 380 now reads as: “Here, we aimed to detect learning about the unintentional social

partner.” Line 392-395 now reads as: “These two templates model intentionality of choices (regardless of outcome), which aims to detect the intentional social interaction, and threat learning (regardless of intentionality), which aims to detect threat expectancy (figure 4 b, “Intentional>Unintentional” and “CS+>CS- “).” Line 416-417 now reads as: “Here, we aimed to detect a gradual learning effect, specifically for CS+_{intentional}.” And line 426-427 as: “Here, we aimed to detect a gradual learning effect, specifically for CS+_{unintentional}.”

2- *The RSA framework can be compared to classification where each pattern is conceived as a “representation” with some information stored. A typical classifier, which is able to discriminate between two categories, is conceptualized in RSA by a similarity matrix where pairs of trials of the same category have high similarity (e.g., intentional-intentional pairs or unintentional-unintentional pairs) and pairs of trials of different category have low similarity (e.g. intentional-unintentional pairs). The proposed regression templates do not follow this logic.*

For example, in the intentional > unintentional template, the authors state that such template predicts higher similarity for intentional pairs of trials than for unintentional ones. However, it is not clear how this relates to a brain regions detecting intentionality vs unintentionality in social interactions. Based on this template (intentional > unintentional), the proposed properties can be summarized as follow: 1) activity patterns during all intentional trials are very similar to each (similarity of 1), as such the same “information” is processed by this brain region for all intentional trials; 2) activity patterns during unintentional trials are anti-correlated (similarity of -1) and each pattern is somewhat unique, as such different “information” is processed by this brain region for each unintentional trials; 3) activity patterns for intentional-unintentional pairs of trials are unrelated (similarity of 0). To my understanding, this template does not implement a brain region capable of detecting intentionality vs unintentionality in social interactions, and I believe the following template would be more appropriate: 1) intentional pairs of trials should exhibit similar patterns (similarity of 1 in the top left triangle); 2) unintentional pairs of trials should also exhibit similar patterns (similarity of 1 in the bottom right triangle); 3)

intentional-unintentional pairs of trials should exhibit different patterns (similarity of 0 in the bottom left quadrant). The same logic should be applied to all templates. In addition to reformulating the templates, I also recommend to further detail the expected neural properties associated with each template in terms of activity patterns and ability to detect/classify different types of stimuli (which is the core of the RSA framework). This will help the reader to understand the relevance of each regression template.

I would like to note that I might be simply missing the relevance of the current models for predicting threat learning in social context, and how this translates into correlated or anti-correlated patterns. In this case, it is important that the authors present these concepts more clearly and explain in details the rationale for their templates.

Authors' response:

We thank the reviewer for this thoroughly written comment and for pointing out a shortcoming of our current analysis method. We agree with the reviewer that the rationale for predicting negative correlations was flawed (confused with contrasts rather than hypothetical data structures) and that such negative correlations would be hard to interpret. To address this shortcoming, and to enhance clarity, we implemented the reviewer's suggestions. We have reconstructed templates with no negative correlations. For instance, the intentionality > unintentionality matrix now contains ones for both intentional pairs and 0 for unintentional pairs (Figure 4b, 2nd from the left), and the CS⁺_{Intentional} > CS_{Other} template now consists of an increase in correlations for the CS⁺_{intentional} trials and 0 for the rest (Figure 4b, 4th matrix from the left). Note that we chose to use an intentionality > unintentionality template which contains 1's for intentional pairs and 0 for unintentional pairs (instead of -1), since we are specifically interested in the effects of intentionality, compared to unintentionality. The novel templates are included in Figure 4 (below), and Electronic Supplementary Material 7-12. The new templates do not alter the pattern of results. We found effects of CS⁺ > CS⁻ in the IFG and the insula during learning, and no effects of intentionality in any of the ROI. These findings are reported in line 570-573 which now reads as: "Category information was represented in the IFG ($t(25) = -4.33$, $p < 0.001$, $d = 0.85$), whereas the insula ($t(25) = 4.02$, $p < 0.001$, $d = 0.78$) and the IFG ($t(25) = 4.31$, $p < 0.001$, $d =$

0.84) (figure 5) showed higher neural pattern similarity for CS+ in contrast to CS- trials.” We refrain from including the exact changes we made to text in this letter, as the resulting changes cover a large area of the text (highlighted in the highlight-edited version of our re-submission).

Figure 4. The RSA regression method. (a). The 28 x 28 RSM for the insula (n = 26), and trial-to-trial correlation values for consecutive trials. (b). Template regression matrices used for the chosen option period. Here, instead of accounting only for the consecutive trials as done in (a), template regression matrices were created to account for similarity between trials that are non-consecutive. As seen in the CS+_{intentional} > CS_{other} template, an increase in correlations was modelled between consecutive trials (off diagonal line) and the rest of the quadrant for the intentional CS+ choices (off diagonal values). (c). The regressor equation used to compute RSA matrix similarity to the regressor templates. Here, templates from (b) are entered into the regressor to assess their individual weights. This allows us to not only test individual contributions of each template but also their relative weights to each other. Compared to the method

used in section (a), this allows for both investigating the off-quadrant values that represent non-consecutive trials, and also to statistically observe the weight each carries.

Reviewer's comment:

3- *I see also other few problematic points with the current templates:*

- *First, from a mathematical perspective, the case of anti-correlated patterns (similarity of -1) is problematic. For example in the intentional > unintentional template, all pairs of unintentional trials have a similarity of -1. Mathematically, it is impossible to conceive 12 different patterns that will all be anti-correlated to each other.*
- *Second, the repeated presentation of the same stimuli is in general expected to lead to rather similar activity patterns for relevant brain regions, so same-category pairs of trials should not have low similarity or anti-correlation (e.g. in the intentional > unintentional template, the same-category unintentional pairs of trials are also set to -1 although the exact same stimuli was presented).*

Authors' response:

We hope that the changes we made based on previous comments have also addressed these concerns. As we removed the negative correlations from the template, we omitted both the mathematical problem of -1 correlation, and the problems related to the interpretation of trial-by-trial anti-correlation. With regard to the second point, previous studies (e.g., Visser et al., 2011, 2013) show that it depends on the brain region whether the repeated presentation of the same stimuli evokes similar activity patterns. While patterns are very consistent in occipital and temporal regions, in frontal and parietal regions they are not. The figure below shows data from the occipital cortex (unpublished figure) next to data from the insula to illustrate this. The color bar presents correlation values between 0 and 0.6. In the occipital cortex correlations within a stimulus category reach .4 (such a structure would be picked up with the category template). In contrast, in the insula, where clear conditioning effects were observed, 'baseline' correlations between stimuli, even between multiple presentations of the same stimulus, are around 0.

Minor comments:

1- *methods, page 15, lines 328-330: “Each matrix consisted of 24*24 elements, each element representing the similarity between the target trials (7*4, 7 trials and 4 conditions)”, 7*4 = 28.*

Authors’ response:

We thank the reviewer for pointing out this typo in the manuscript. Line 331 now read as: “Each matrix consisted of 28*28 elements...”

2- *methods, page 21, lines 449-452: the group-level statistics should be performed on the individual weights rather than the associated individual t-values (as in univariate GLM analyses where beta estimates from the single subject level are fed into the group level GLM).*

Authors’ response:

We thank the reviewer for this suggestion. This change has been implemented, and we have updated the analysis to use estimates from the regression instead of t-values for the group statistics. Lines 445-449 now read as following: “For each subject and each ROI, we regressed template regression matrices to RSM’s, resulting in weights representing how much each of the template regression matrices was able to capture

each RSM. To assess if we had any effects in the whole participant sample, we ran a one-sample t-test on these estimates to see if they significantly differed from zero across the sample.” Line 460-465 now reads: “Second, for each model, ROI, and coefficient, the distributions of the beta estimates from all participants were tested against zero using a one-sample t-test.”. We also report the distribution of the estimates in Figure 5, instead of t-values, as seen below (Line 646 in the manuscript):

Figure 5. RSM’s and regression results (n = 26). Distribution of regression estimates for all participants, for each template regression matrix, in the IFG and insula. *** $p < 0.001$, FDR-corrected.

3- *methods, page 21, lines 455-457: If the trial-by-trial correlation matrices are Fischer-transformed, I believe the regression templates should also be Fischer-transformed for the regression.*

Authors’ response:

We thank the reviewer for suggesting this change. We implemented this change as suggested, and Fisher transformed the template matrices. Line 452-453 now reads as: “Each regressor matrix was Fisher-transformed. Since perfect correlation values return -Infinity or Infinity in such a transformation, we replaced these infinity values with 0.999 after the transformation.”

4- *results, pages 28-30: it is not clear whether the reported ROIs with significant fitting were found in the right or left hemispheres, please specify.*

Authors' response:

Line 306 now reads as: "Masks for all ROIs, apart from the left and right TPJ, were applied bilaterally."

5- figure 3: the bar charts and legends have different sizes across the different panels.

Authors' response:

We thank the reviewer for pointing out the difference in sizes. We have adjusted the figure to have a similar size for all charts, as seen below.

References:

1. Visser RM, Scholte HS, Kindt M. Associative Learning Increases Trial-by-Trial Similarity of BOLD-MRI Patterns. *J Neurosci.* 2011;31(33):12021-12028. doi:10.1523/JNEUROSCI.2178-11.2011
2. Visser RM, Scholte HS, Beemsterboer T, Kindt M. Neural pattern similarity predicts long-term fear memory. *Nat Neurosci.* 2013;16(4):388-390. doi:10.1038/nn.3345

Appendix C

Dear Prof. Viding,

Thank you for your positive decision. We were thrilled to learn that our manuscript was accepted with minor revision, and thankful for the final comments from yourself and Reviewer 3, all of which we think will further enhance the quality of our paper. Here, below, please find our responses to yours and the reviewer's comments.

Reviewer: 3

Comments to the Author(s)

This is the second time I review this manuscript. The authors have satisfactorily addressed my main concerns. I have a couple of remaining minor issues that should be addressed/discussed.

Authors' response:

We would like to thank the reviewer for the constructive feedback. We hope that the reviewer finds the manuscript improved.

minor concerns:

- *One of the main topic addressed by the authors is the context of intentional vs unintentional social interactions. They could not replicate previous results about intentionality and only wrote: "Although we replicated the behavioural effects of intentionality (1), we did not observe the expected neural effects, which might be related to a lack of power and/or methodological differences as compared to previous work." The authors should develop this paragraph to provide better explanations about the lack of results for the intentionality effect, which is conceived as a major aspect of the experimental design.*

Authors' response:

We have now developed this paragraph, which now reads as follows (line 735 - 743): "Unlike the CS+, the intentionality manipulation was not accompanied by a reminder cue (e.g. reminding the participant about the intentionality of the current trial). The lack of a reminder might have contributed to a weaker effect, especially in comparison to the conditioning effect. Indeed, to capture the effects of intentionality in the presence

of a salient stimulus like the CS+, a greater sample size might be necessary. Our previous study (Undeger, Visser, & Olsson, 2020), where we were able to capture the effects of intentionality in fear learning, contained a greater sample size. Future research could include reminders about the intentionality and/or a larger sample size.”

• A possible way to address the lack of conclusive results about intentionality would be to explore other ROIs. In the present and previous work from the authors, they select a priori ROIs based on the literature. I would recommend exploring a bit more the data by applying a wholebrain approach to the RSA or by using the same data with a univariate GLM approach to identify the ROIs in which the BOLD signal varies significantly as a function of intentionality and to further apply RSA within these ROIs (no problem of double dipping with univariate and multivariate analyses applied to the same data). Although restricting the number of tested ROIs is in principle a good approach, I believe in this case, with a novel paradigm, a more explorative approach could be justified.

Authors' response:

As suggested by the Reviewer, we conducted a whole-brain GLM. Contrasting CS⁺_{intentional} and CS⁺_{unintentional} showed activity in visual areas, the left TPJ (figure 1 below), and the frontal pole and parietal areas (table 1 below). These clusters included the insula (figure 1, upper right panel), and the dmPFC (figure 1, upper left panel). We included target events and 6 motion parameters to conduct the analyses. Each GLM included target events, and 6 motion parameters. All lower-level analyses have been conducted using a voxel threshold of $z > 2.3$, and $p < 0.05$. All higher-level analyses were conducted with a cluster threshold of $z > 3.1$, $p < 0.05$ using the mixed model, applying FLAME 1 (FMRIB's local analysis of mixed effects).

We report our findings in the main text; lines 598-607 now reads: “We used a univariate GLM approach to explore intentionality related activity in the brain regions that were not included in our initial set of ROIs. We modelled the target trials, the filler trials, the anticipation period, the chosen option period, and US delivery. For a more detailed account of the results and the methods of the univariate analyses, please see

electronic supplementary material 14-16. Although, we did not find a main effect of intentionality (Intentional > Unintentional), the $CS^+_{\text{intentional}} > CS^+_{\text{unintentional}}$, engaged the insula, the dmPFC and the rTPJ (see electronic supplementary material 14, 15). The effects in the insula were also present for the $CS^+ > CS^-$ (i.e. CS^+ activity regardless of intentionality) (electronic supplementary material 14, 16). The contrast $CS^+_{\text{unintentional}} > CS^+_{\text{intentional}}$ yielded no significant clusters of activity.”

Figure 1: Whole-brain activation for the integration of intentionality and CS^+ information. In red-yellow, $CS^+_{\text{intent}} > CS^+_{\text{unintent}}$ contrast, in blue the $CS^+ > CS^-$ contrast.

Region	X	Y	Z	Voxels	Z
L LOC, inf	-38	-84	-12	43743	7.65
L TPJ	-58	-36	22	1160	5.31
L PG	-54	0	48	682	5.26
L FP	-40	38	30	440	4.88
R SPL	18	-46	66	195	4.44
R FP	30	58	-4	149	4.3

All values are z-thresholded at 3.1. Anatomical labels are based on the Harvard-Oxford Structural Atlas. LOC = Lateral Occipital Cortex, inferior division; TPJ = Temporo-parietal Junction; PG = Precentral Gyrus, FP = Frontal Pole, SPL = Superior Parietal Lobule.

Table 1: Univariate activation $CS_{+intent} > CS_{+unintent}$.